# Joint Embedding Self-Supervised Learning in the Kernel Regime

## Abstract

The fundamental goal of self-supervised learning (SSL) is to produce useful representations of data without access to any labels for classifying the data. Modern methods in SSL, which form representations based on known or constructed relationships between samples, have been particularly effective at this task. Motivated by a rich line of work in kernel methods performed on graphs and manifolds, we show that SSL methods likewise admit a kernel regime where embeddings are constructed by linear maps acting on the feature space of a kernel to find the optimal form of the output representations for contrastive and non-contrastive loss functions. This procedure produces a new representation space with an inner product denoted as the induced kernel which generally correlates points which are related by an augmentation in kernel space and de-correlates points otherwise. We analyze our kernel model on small datasets to identify common features of self-supervised learning algorithms and gain theoretical insights into their performance on downstream tasks.

## 1 Introduction

Self-supervised learning (SSL) algorithms are broadly tasked with learning from unlabeled data. In the joint embedding framework of SSL, mainstream contrastive methods build representations by reducing the distance between inputs related by an augmentation (positive pairs) and increasing the distance between inputs not known to be related (negative pairs) (Chen et al., 2020; He et al., 2020; Oord et al., 2018; Ye et al., 2019). Non-contrastive methods only enforce similarities between positive pairs but are designed carefully to avoid collapse of representations (Grill et al., 2020; Zbontar et al., 2021). Recent algorithms for SSL have performed remarkably well reaching similar performance to baseline supervised learning algorithms on many downstream tasks (Caron et al., 2020; Bardes et al., 2021; Chen & He, 2021).

In this work, we study SSL from a kernel perspective motivated by the rich history of study of kernel algorithms on graphs and manifolds (Smola & Kondor, 2003; Ando & Zhang, 2006; Bellet et al., 2013; Belkin & Niyogi, 2004). Our primary aim is to extend this line of work to cover commonly used loss functions in modern SSL settings potentially providing useful insights into their properties and performance. In standard SSL tasks, inputs are fed into a neural network and mapped into a feature space which encodes the final representations used in downstream tasks (e.g., classification tasks). In the kernel setting, inputs are embedded in a feature space corresponding to a kernel, and representations are constructed via an optimal mapping from this feature space to the vector space for the representations of the data. Here, the task can be framed as one of finding an optimal "induced" kernel, which is a mapping from the original kernel in the input feature space to an updated kernel function acting on the vector space of the representations. Our results show that such an induced kernel can be constructed using only manipulations of kernel functions and data that encodes the relationships between inputs in an SSL algorithm (e.g., adjacency matrices between the input datapoints).

More broadly, we make the following contributions:

- For a contrastive and non-contrastive loss, we provide closed form solutions when the algorithm is trained over a single batch of data. These solutions form a new "induced" kernel which can be used to perform downstream supervised learning tasks.

- We show that a version of the representer theorem in kernel methods can be used to formulate kernelized SSL tasks as optimization problems. As an example, we show how to optimally find induced kernels when the loss is enforced over separate batches of data.

- We empirically study the properties of our SSL kernel algorithms to gain insights about the training of SSL algorithms in practice. We study the generalization properties of SSL algorithms and show that the choice of augmentation and adjacency matrices encoding relationships between the datapoints are crucial to performance.

We proceed as follows. First, we provide a brief background of the goals of our work and the theoretical tools used in our study. Second, we show that kernelized SSL algorithms trained on a single batch admit a closed form solution for commonly used contrastive and non-contrastive loss functions. Third, we generalize our findings to provide a semi-definite programming formulation to solve for the optimal induced kernel in more general settings and provide heuristics to better understand the form and properties of the induced kernels. Finally, we empirically investigate our kernelized SSL algorithms when trained on various datasets (code included in supplemental material).

## 1.1 NOTATION AND SETUP

We denote vectors and matrices with lowercase ($\boldsymbol{x}$) and uppercase ($\boldsymbol{X}$) letters respectively. The vector 2-norm and matrix operator norm is denoted by $\| \cdot \|$. The Frobenius norm of a matrix $\boldsymbol{M}$ is denoted as $\|\boldsymbol{M}\|_F$. We denote the transpose and conjugate transpose of $\boldsymbol{M}$ by $\boldsymbol{M}^\intercal$ and $\boldsymbol{M}^\dagger$ respectively. We denote the identity matrix as $\boldsymbol{I}$ and the vector with each entry equal to one as $\boldsymbol{1}$. For a diagonalizable matrix $\boldsymbol{M}$, its projection onto the eigenspace of its positive eigenvalues is $\boldsymbol{M}_+$.

For a dataset of size $N$, let $\boldsymbol{x}_i \in \mathcal{X}$ for $i \in [N]$ denote the elements of the dataset. Given a kernel function $k : \mathcal{X} \times \mathcal{X} \to \mathbb{R}$, let $\Phi(\boldsymbol{x}) = k(\boldsymbol{x}, \cdot)$ be the map from inputs to the reproducing kernel Hilbert space (RKHS) denoted by $\mathcal{H}$ with corresponding inner product $\langle \cdot, \cdot \rangle_{\mathcal{H}}$ and RKHS norm $\| \cdot \|_{\mathcal{H}}$. Throughout we denote $\boldsymbol{K}_{s,s} \in \mathbb{R}^{N \times N}$ to be the kernel matrix of the SSL dataset where $(\boldsymbol{K}_{s,s})_{ij} = k(\boldsymbol{x}_i, \boldsymbol{x}_j)$. We consider linear models $\boldsymbol{W} : \mathcal{H} \to \mathbb{R}^K$ which map features to representations $\boldsymbol{z}_i = \boldsymbol{W}\Phi(\boldsymbol{x}_i)$. Let $\boldsymbol{Z}$ be the representation matrix which contains $\Phi(\boldsymbol{x}_i)$ as rows of the matrix. This linear function space induces a corresponding RKHS norm which can be calculated as $\|\boldsymbol{W}\|_{\mathcal{H}} = \sqrt{\sum_{i=1}^{K} \langle \boldsymbol{W}_i, \boldsymbol{W}_i \rangle_{\mathcal{H}}^2}$ where $\boldsymbol{W}_i \in \mathcal{H}$ denotes the $i$-th component of the output of the linear mapping $\boldsymbol{W}$. This linear mapping constructs an "induced" kernel denoted as $k^*(\cdot, \cdot)$ as discussed later.

The driving motive behind modern self-supervised algorithms is to maximize the information of given inputs in a dataset while enforcing similarity between inputs that are known to be related. The adjacency matrix $\boldsymbol{A} \in \{0, 1\}^{N \times N}$ (also can be generalized to $\boldsymbol{A} \in \mathbb{R}^{N \times N}$) connects related inputs $\boldsymbol{x}_i$ and $\boldsymbol{x}_j$ (i.e., $\boldsymbol{A}_{ij} = 1$ if inputs $i$ and $j$ are related by a transformation) and $\boldsymbol{D_A}$ is a diagonal matrix where entry $i$ on the diagonal is equal to the number of nonzero elements of row $i$ of $\boldsymbol{A}$.

## 2 RELATED WORKS

In this section, we briefly summarize some of the related works for this study. We include a more detailed related works section in Appendix A.

**Modern self-supervised learning approaches:** Joint embedding approaches to SSL produce representations by comparing representations of inputs via known relationships. Methods are denoted as non-contrastive if the loss function is only a function of pairs that are related (Grill et al., 2020; Chen & He, 2021; Zbontar et al., 2021; Bardes et al., 2021). Contrastive methods also penalize similarities of representations that are not related. Popular algorithms include SimCLR (Chen et al., 2020), SwAV (Caron et al., 2020), NNCLR (Dwibedi et al., 2021), contrastive predictive coding (Oord et al., 2018), spectral contrastive loss (HaoChen et al., 2021), and many others. Separate from the joint embedding framework, many methods in SSL form representations by predicting held-out portions of the data typically in a generative model setting. Commonly used algorithms incorporate autoencoder approaches (He et al., 2022; Radford et al., 2019; Vincent et al., 2010; Dosovitskiy et al., 2020) which are used in both natural language processing and image processing tasks.

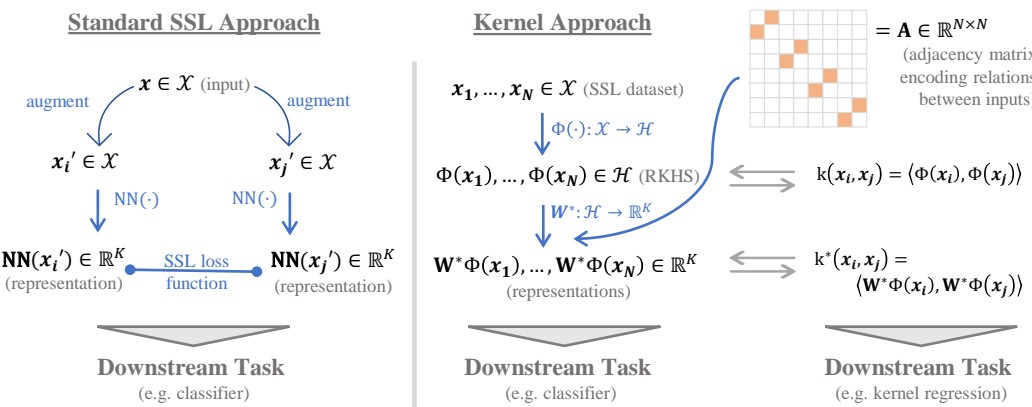

Figure 1: To translate the SSL setting into the kernel regime, we aim to find the optimal linear function which maps inputs from the RKHS into the $K$-dimensional feature space of the representations. This new feature space induces a new optimal kernel denoted the "induced" kernel. Relationships between data-points are encoded in an adjacency matrix (the example matrix shown here contains pairwise relationships between datapoints).

**Neural tangent kernels and gaussian processes:** Prior work has connected the outputs of infinite width neural networks to a corresponding gaussian process (Williams & Rasmussen, 2006; Neal, 1996; Lee et al., 2017). When trained using continuous time gradient descent, these infinite width models evolve as linear models under the so called neural tangent kernel (NTK) regime (Jacot et al., 2018; Arora et al., 2019a). The discovery of the NTK opened a flurry of exploration into the connections between so-called lazy training of wide networks and kernel methods (Yang, 2019; Chizat & Bach, 2018; Wang et al., 2022; Bietti & Mairal, 2019). Though the training dynamics of the NTK has previously been studied in the supervised settings, one can analyze an NTK in a self-supervised setting by using that kernel in the SSL algorithms that we study here. We perform some preliminary investigation into this direction in our experiments.

**Kernel and metric learning:** Various works have proposed a series of graph-based kernels that have ties to semi-supervised learning and appealing regularization properties over the geometric structure of the data (Smola & Kondor, 2003; Belkin et al., 2006; Vishwanathan et al., 2010). Links between spectral properties of the graphs and representations are often formed during the process of learning (Belkin & Niyogi, 2004; Zhao & Liu, 2012). Throughout our study, we note connections to these related works. From an algorithmic perspective, perhaps the closest lines of work are related to kernel and metric learning (Bellet et al., 2013; Yang & Jin, 2006). Since our focus is on directly kernelizing SSL methods to eventually analyze and better understand SSL algorithms, our end goal is not to improve over these methods but instead, to extend them to the SSL setting. These works are further summarized in Appendix A.

## 3 CONTRASTIVE AND NON-CONTRASTIVE KERNEL METHODS

Stated informally, the goal of SSL in the kernel setting is to start with a given kernel function $k : \mathcal{X} \times \mathcal{X} \to \mathbb{R}$ (e.g., RBF kernel or a neural tangent kernel) and map this kernel function to a new "induced" kernel $k^* : \mathcal{X} \times \mathcal{X} \to \mathbb{R}$ which is a function of the SSL loss function and the SSL dataset. For two new inputs $x$ and $x'$, the induced kernel $k^*(x, x')$ generally outputs correlated values if $x$ and $x'$ are correlated in the original kernel space to some datapoint in the SSL dataset or correlated to separate but related datapoints in the SSL dataset as encoded in the graph adjacency matrix. If no relations are found in the SSL dataset between $x$ and $x'$, then the induced kernel will generally output an uncorrelated value.

To kernelize SSL methods, we consider a setting generalized from the prototypical SSL setting where representations are obtained by maximizing/minimizing distances between augmented/unaugmented samples. Translating this to the kernel regime, as illustrated in Figure 1, our goal is to find a linear mapping $W^* : \mathcal{H} \to \mathbb{R}^K$ which obtains the optimal representation of the data for a

given SSL loss function and minimizes the RKHS norm. This optimal solution produces an "induced kernel" $k^*(\cdot, \cdot)$ which is the inner product of the data in the output representation space. Once constructed, the induced kernel can be used in downstream tasks to perform supervised learning.

Due to a generalization of the representer theorem (Schölkopf et al., 2001), we can show that the optimal linear function $\boldsymbol{W}^*$ must be in the support of the data. This implies that the induced kernel can be written as a function of the kernel between datapoints in the SSL dataset.

**Proposition 3.1** (Form of optimal representation). *Given a dataset $\boldsymbol{x}_1, \ldots, \boldsymbol{x}_N \in \mathcal{X}$, let $k(\cdot, \cdot)$ be a kernel function with corresponding map $\Phi : \mathcal{X} \to \mathcal{H}$ into the RKHS $\mathcal{H}$. Let $\boldsymbol{W} : \mathcal{H} \to \mathbb{R}^K$ be a function drawn from the space of linear functions $\mathcal{W}$ mapping inputs in the RKHS to the vector space of the representation. For a risk function $\mathcal{R}(\boldsymbol{W}\Phi(\boldsymbol{x}_1), \ldots, \boldsymbol{W}\Phi(\boldsymbol{x}_N)) \in \mathbb{R}$ and any strictly increasing function $r : [0, \infty) \to \mathbb{R}$, consider the optimization problem*

$$\boldsymbol{W}^* = \arg\min_{\boldsymbol{W} \in \mathcal{W}} \mathcal{R}(\boldsymbol{W}\Phi(\boldsymbol{x}_1), \ldots, \boldsymbol{W}\Phi(\boldsymbol{x}_N)) + r\left(\|\boldsymbol{W}\|_{\mathcal{H}}\right). \tag{1}$$

*The optimal solutions of the above take the form*

$$\begin{aligned}\text{optimal representation: } & \boldsymbol{W}^*\Phi(\boldsymbol{x}) = \boldsymbol{M}\boldsymbol{k}_{\boldsymbol{X}, \boldsymbol{x}} \\ \text{induced kernel: } & k^*(\boldsymbol{x}, \boldsymbol{x}') = (\boldsymbol{M}\boldsymbol{k}_{\boldsymbol{X}, \boldsymbol{x}})^{\mathsf{T}} \boldsymbol{M}\boldsymbol{k}_{\boldsymbol{X}, \boldsymbol{x}'} = \boldsymbol{k}_{\boldsymbol{X}, \boldsymbol{x}}^{\mathsf{T}} \boldsymbol{M}^{\mathsf{T}} \boldsymbol{M}\boldsymbol{k}_{\boldsymbol{X}, \boldsymbol{x}'},\end{aligned} \tag{2}$$

*where $\boldsymbol{M} \in \mathbb{R}^{K \times N}$ is a matrix that must be solved for and $\boldsymbol{k}_{\boldsymbol{X}, \boldsymbol{x}} \in \mathbb{R}^N$ is a vector with entries $[\boldsymbol{k}_{\boldsymbol{X}, \boldsymbol{x}}]_i = k(\boldsymbol{x}_i, \boldsymbol{x})$.*

Proposition 3.1, proved in Appendix B.1, provides a prescription for finding the optimal representations or induced kernels: i.e, one must search over the set of matrices $\boldsymbol{M} \in \mathbb{R}^{K \times N}$ to find an optimal matrix. This search can be performed using standard optimization techniques as we will discuss later, but in certain cases, the optimal solution can be calculated in closed-form as shown next for both a contrastive and non-contrastive loss function.

**Non-contrastive loss** Consider a variant of the VICReg (Bardes et al., 2021) loss function below:

$$\mathcal{L}_{VIC} = \left\| \boldsymbol{Z}^{\mathsf{T}} \left( \boldsymbol{I} - \frac{1}{N}\boldsymbol{1}\boldsymbol{1}^{\mathsf{T}} \right) \boldsymbol{Z} - \boldsymbol{I} \right\|_F^2 + \beta \operatorname{Tr}\left[\boldsymbol{Z}^{\mathsf{T}}\boldsymbol{L}\boldsymbol{Z}\right], \tag{3}$$

where $\beta \in \mathbb{R}^+$ is a hyperparameter that controls the invariance term in the loss and $\boldsymbol{L} = \boldsymbol{D}_{\boldsymbol{A}} - \boldsymbol{A}$ is the graph Laplacian of the data. We note that the second term in the above loss function takes a role akin to the Laplacian or manifold regularization term studied in kernel methods over graphs (Ando & Zhang, 2006; Belkin et al., 2006). When the representation space has dimension $K \geq N$ and the kernel matrix of the data is full rank, the induced kernel of the above loss function is:

$$k^*(\boldsymbol{x}, \boldsymbol{x}') = \boldsymbol{k}_{\boldsymbol{x}, s}\boldsymbol{K}_{s,s}^{-1} \left( \boldsymbol{I} - \frac{1}{N}\boldsymbol{1}\boldsymbol{1}^{\mathsf{T}} - \frac{\beta}{2}\boldsymbol{L} \right)_+ \boldsymbol{K}_{s,s}^{-1}\boldsymbol{k}_{s,\boldsymbol{x}'}, \tag{4}$$

where $(\cdot)_+$ projects the matrix inside the parentheses onto the eigenspace of its positive eigenvalues, $\boldsymbol{k}_{\boldsymbol{x}, s} \in \mathbb{R}^{1 \times N}$ is the kernel row-vector with entry $i$ equal to $k(\boldsymbol{x}, \boldsymbol{x}_i)$ with $\boldsymbol{k}_{s, \boldsymbol{x}}$ equal to its transpose, and $\boldsymbol{K}_{s,s} \in \mathbb{R}^{N \times N}$ is the kernel matrix of the training data for the self-supervised dataset where entry $i, j$ is equal to $k(\boldsymbol{x}_i, \boldsymbol{x}_j)$. When we restrict the output space of the self-supervised learning task to be of dimension $K < N$, then the induced kernel only incorporates the top $K$ eigenvectors of $\boldsymbol{I} - \frac{1}{N}\boldsymbol{1}\boldsymbol{1}^{\mathsf{T}} - \frac{\beta}{2}\boldsymbol{L}$:

$$k^*(\boldsymbol{x}, \boldsymbol{x}') = \boldsymbol{k}_{\boldsymbol{x}, s}\boldsymbol{K}_{s,s}^{-1}\boldsymbol{C}_{:, \leq K}\boldsymbol{D}_{\leq K, \leq K}\boldsymbol{C}_{:, \leq K}^{\mathsf{T}}\boldsymbol{K}_{s,s}^{-1}\boldsymbol{k}_{s, \boldsymbol{x}'}, \tag{5}$$

where $\boldsymbol{C}\boldsymbol{D}\boldsymbol{C}^{\mathsf{T}} = \boldsymbol{I} - \frac{1}{N}\boldsymbol{1}\boldsymbol{1}^{\mathsf{T}} - \frac{\beta}{2}\boldsymbol{L}$ is the eigendecomposition including only positive eigenvalues sorted in descending order, $\boldsymbol{C}_{:, \leq K}$ denotes the matrix consisting of the first $K$ columns of $\boldsymbol{C}$ and $\boldsymbol{D}_{\leq K, \leq K}^{1/2}$ denotes the $K \times K$ matrix consisting of entries in the first $K$ rows and columns. Proofs of the above are in Appendix B.2.

**Contrastive loss** For contrastive SSL, we can also obtain a closed form solution to the induced kernel for a variant of the spectral contrastive loss (HaoChen et al., 2021):

$$\mathcal{L}_{sc} = \|\boldsymbol{Z}\boldsymbol{Z}^{\mathsf{T}} - (\boldsymbol{I} + \boldsymbol{A})\|_F^2, \tag{6}$$

where $\boldsymbol{A}$ is the adjacency matrix encoding relations between datapoints. When the representation space has dimension $K \geq N$, this loss results in the optimal induced kernel:

$$k^*(\boldsymbol{x}, \boldsymbol{x}') = \boldsymbol{k}_{\boldsymbol{x},s} \boldsymbol{K}_{s,s}^{-1} (\boldsymbol{I} + \boldsymbol{A})_+ \boldsymbol{K}_{s,s}^{-1} \boldsymbol{k}_{s,\boldsymbol{x}'}, \tag{7}$$

where $(\boldsymbol{I} + \boldsymbol{A})_+$ is equal to the projection of $\boldsymbol{I} + \boldsymbol{A}$ onto its eigenspace of positive eigenvalues. In the standard SSL setting where relationships are pair-wise (i.e., $\boldsymbol{A}_{ij} = 1$ if $\boldsymbol{x}_i$ and $\boldsymbol{x}_j$ are related by an augmentation), then $\boldsymbol{I} + \boldsymbol{A}$ has only positive or zero eigenvalues so the projection can be ignored. If $K < N$, then we similarly project the matrix $\boldsymbol{I} + \boldsymbol{A}$ onto its top $K$ eigenvalues and obtain an induced kernel similar to the non-contrastive one:

$$k^*(\boldsymbol{x}, \boldsymbol{x}') = \boldsymbol{k}_{\boldsymbol{x},s} \boldsymbol{K}_{s,s}^{-1} \boldsymbol{C}_{:,\leq K} \boldsymbol{D}_{\leq K, \leq K} \boldsymbol{C}_{:,\leq K}^{\mathsf{T}} \boldsymbol{K}_{s,s}^{-1} \boldsymbol{k}_{s,\boldsymbol{x}'}, \tag{8}$$

where as before, $\boldsymbol{C}\boldsymbol{D}\boldsymbol{C}^{\mathsf{T}} = \boldsymbol{I} + \boldsymbol{A}$ is the eigendecomposition including only positive eigenvalues with eigenvalues in descending order, $\boldsymbol{C}_{:,\leq K}$ consists of the first $K$ columns of $\boldsymbol{C}$ and $\boldsymbol{D}_{\leq K, \leq K}^{1/2}$ is the $K \times K$ matrix of the first $K$ rows and columns. Proofs of the above are in Appendix B.3. Note, that the induced kernels generally correlate data over the top eigenvalues of the graph adjacency matrix in line with findings from previous works in kernel methods and spectral graph theory (Zhao & Liu, 2007; Belkin & Niyogi, 2004).

## 3.1 General form as SDP

The closed form solutions for the induced kernel obtained above assumed the loss function was enforced across a single batch. Of course, in practice, data are split into several batches. This batched setting may not admit a closed-form solution, but by using Proposition 3.1, we know that any optimal induced kernel takes the general form:

$$k^*(\boldsymbol{x}, \boldsymbol{x}') = \boldsymbol{k}_{\boldsymbol{x},s} \boldsymbol{B} \boldsymbol{k}_{s,\boldsymbol{x}'}, \tag{9}$$

where $\boldsymbol{B} \in \mathbb{R}^{N \times N}$ is a positive semi-definite matrix. With constraints properly chosen so that the solution for each batch is optimal (Balestriero & LeCun, 2022; HaoChen et al., 2022), one can find the optimal matrix $\boldsymbol{B}^*$ by solving a semi-definite program (SDP). We perform this conversion to a SDP for the contrastive loss here and leave proofs and further details including the non-contrastive case to Appendix B.4.

Introducing some notation to deal with batches, assume we have $N$ datapoints split into $n_{\text{batches}}$ of size $b$. We denote the $i$-th datapoint within batch $j$ as $\boldsymbol{x}_i^{(j)}$. As before, $\boldsymbol{x}_i$ denotes the $i$-th datapoint across the whole dataset. Let $\boldsymbol{K}_{s,s} \in \mathbb{R}^{N \times N}$ be the kernel matrix over the complete dataset where $[\boldsymbol{K}_{s,s}]_{i,j} = k(\boldsymbol{x}_i, \boldsymbol{x}_j)$, $\boldsymbol{K}_{s,s_j} \in \mathbb{R}^{N \times b}$ be the kernel matrix between the complete dataset and batch $j$ where $[\boldsymbol{K}_{s,s_j}]_{a,b} = k(\boldsymbol{x}_a, \boldsymbol{x}_b^{(j)})$, and $\boldsymbol{A}^{(j)}$ be the adjacency matrix for inputs in batch $j$. With this notation, we now aim to minimize the loss function adapted from Equation (6) including a regularizing term for the RKHS norm:

$$\mathcal{L} = \sum_{j=1}^{n_{\text{batches}}} \left\| \boldsymbol{K}_{s_j,s} \boldsymbol{B} \boldsymbol{K}_{s,s_j} - \left( \boldsymbol{I} + \boldsymbol{A}^{(j)} \right) \right\|_F^2 + \alpha \operatorname{Tr}(\boldsymbol{B} \boldsymbol{K}_{s,s}), \tag{10}$$

where $\alpha \in \mathbb{R}^+$ is a weighting term for the regularizer. Taking the limit of $\alpha \to 0$, we can find the optimal induced kernel for a representation of dimension $K > b$ by enforcing that optimal representations are obtained in each batch:

$$\min_{\boldsymbol{B} \in \mathbb{R}^{N \times N}} \operatorname{Tr}(\boldsymbol{B} \boldsymbol{K}_{s,s})$$
$$\text{s.t. } \boldsymbol{K}_{s_j,s} \boldsymbol{B} \boldsymbol{K}_{s,s_j} = \left( \boldsymbol{I} + \boldsymbol{A}^{(j)} \right)_+ \quad \forall j \in \{1, 2, \ldots, n_{\text{batches}}\} \tag{11}$$
$$\boldsymbol{B} \succeq 0, \ \operatorname{rank}(\boldsymbol{B}) = K,$$

where as before, where $(\boldsymbol{I} + \boldsymbol{A}^{(j)})_+$ is equal to the projection of $\boldsymbol{I} + \boldsymbol{A}^{(j)}$ onto its eigenspace of positive eigenvalues. Relaxing and removing the constraint that $\operatorname{rank}(\boldsymbol{B}) = K$ results in an SDP which can be efficiently solved using existing optimizers. Further details and a generalization of this conversion to other representation dimensions is shown in Appendix B.4.

## 3.2 INTERPRETING THE INDUCED KERNEL

The top eigenvectors of the adjacency matrix form the representations in the SSL tasks studied here and are consistent with related approaches in kernel methods and spectral graph theory (Zhao & Liu, 2007; Belkin & Niyogi, 2004). More generally, as a loose rule, the induced kernel will correlate points that are close in the kernel space or related by augmentations in the SSL dataset and uncorrelate points otherwise. Stated in the framework of (Wang & Isola, 2020), the induced kernel increases alignment by enforcing correlation in datapoints in the traning set and achieves uniformity by setting elements of the representation space to be orthogonal eigenvectors over the dataset. As an example, note that in the contrastive setting (Equation (7)), if one calculates the induced kernel $k^*(\boldsymbol{x}_i, \boldsymbol{x}_j)$ between two points in the SSL dataset indexed by $i$ and $j$ that are related by an augmentation (i.e., $\boldsymbol{A}_{ij} = 1$), then the kernel between these two points is $k^*(\boldsymbol{x}_i, \boldsymbol{x}_j) = 1$. More generally, if the two inputs to the induced kernel are close in kernel space to different points in the SSL dataset that are known to be related by $\boldsymbol{A}$, then the kernel value will be close to 1. We formalize this intuition below for the standard setting with pairwise augmentations.

**Proposition 3.2.** *Given kernel function $k(\cdot, \cdot)$ with corresponding map $\Phi(\cdot)$ into the RKHS $\mathcal{H}$, let $\{\boldsymbol{x}_1, \boldsymbol{x}_2, \ldots \boldsymbol{x}_N\}$ be an SSL dataset normalized such that $k(\boldsymbol{x}_i, \boldsymbol{x}_i) = 1$ and formed by pairwise augmentations (i.e., every element has exactly one neighbor in $\boldsymbol{A}$) with kernel matrix $\boldsymbol{K}_{s,s}$. Given two points $\boldsymbol{x}$ and $\boldsymbol{x}'$, if there exists two points in the SSL dataset indexed by $i$ and $j$ which are related by an augmentation ($\boldsymbol{A}_{ij}=1$) and $\|\Phi(\boldsymbol{x}) - \Phi(\boldsymbol{x}_i)\|_{\mathcal{H}} \leq \frac{\Delta}{5\|\|\boldsymbol{K}_{s,s}^{-1}\|\sqrt{N}}$ and $\|\Phi(\boldsymbol{x}') - \Phi(\boldsymbol{x}_j)\|_{\mathcal{H}} \leq \frac{\Delta}{5\|\|\boldsymbol{K}_{s,s}^{-1}\|\sqrt{N}}$, then the induced kernel for the contrastive loss is at least $k^*(\boldsymbol{x}, \boldsymbol{x}') \geq 1 - \Delta$.*

We prove the above statement in Appendix B.5. The bounds in the above statement which depend on the number of datapoints $N$ and the kernel matrix norm $\|\boldsymbol{K}_{s,s}^{-1}\|$ are not very tight and solely meant to provide intuition for the properties of the induced kernel. In more realistic settings, stronger correlations will be observed for much weaker values of the assumptions. In light of this, we visualize the induced kernel values and their relations to the original kernel function in Section 4.1 on a simple 2-dimensional spiral dataset. Here, it is readily observed that the induced kernel better connects points along the data manifold that are related by the adjacency matrix.

## 3.3 DOWNSTREAM TASKS

In downstream tasks, one can apply the induced kernels directly on supervised algorithms such as kernel regression or SVM. Alternatively, one can also extract representations directly by obtaining the representation as $k^*(\boldsymbol{x}, \cdot) = \boldsymbol{M}\boldsymbol{k}_{s,\boldsymbol{x}}$ as shown in Proposition 3.1 and employ any learning algorithm from these features. As an example, in kernel regression, we are given a dataset of size $N_t$ consisting of input-output pairs $\{\boldsymbol{x}_i^{(t)}, y_i\}$ and aim to train a linear model to minimize the mean squared error loss of the outputs (Williams & Rasmussen, 2006). The optimal solution using an induced kernel $k^*(\cdot, \cdot)$ takes the form:

$$f^*(\boldsymbol{x}) = \left[ k^*(\boldsymbol{x}, \boldsymbol{x}_1^{(t)}), k^*(\boldsymbol{x}, \boldsymbol{x}_2^{(t)}), \ldots, k^*(\boldsymbol{x}, \boldsymbol{x}_{N_t}^{(t)}) \right] \cdot \boldsymbol{K}_{t,t}^{*-1} \boldsymbol{y}, \tag{12}$$

where $\boldsymbol{K}_{t,t}^{*-1}$ is the kernel matrix of the supervised training dataset with entry $i, j$ equal to $k^*(\boldsymbol{x}_i^{(t)}, \boldsymbol{x}_j^{(t)})$ and $\boldsymbol{y}$ is the concatenation of the targets as a vector. Note that since kernel methods generally have complexity that scales quadratically with the number of datapoints, such algorithms may be unfeasible in large-scale learning tasks unless modifications are made.

A natural question is when and why should one prefer the induced kernel of SSL to a kernel used in the standard supervised setting perhaps including data augmentation? Kernel methods generally fit a dataset perfectly so an answer to the question more likely arises from studying generalization. In kernel methods, generalization error typically tracks with the norm of the classifier captured by the complexity quantity $s_N(\boldsymbol{K})$ defined as (Mohri et al., 2018; Steinwart & Christmann, 2008):

$$s_N(\boldsymbol{K}) = \frac{\text{Tr}(\boldsymbol{K})}{N} \boldsymbol{y}^{\intercal} \boldsymbol{K}^{-1} \boldsymbol{y}, \tag{13}$$

where $\boldsymbol{y}$ is a vector of targets and $\boldsymbol{K}$ is the kernel matrix of the supervised dataset. For example, the generalization gap of an SVM algorithm can be bounded with high probability by $O(\sqrt{s_N(\boldsymbol{K})/N})$ (see example proof in Appendix C.1) (Meir & Zhang, 2003; Huang et al., 2021). For kernel functions

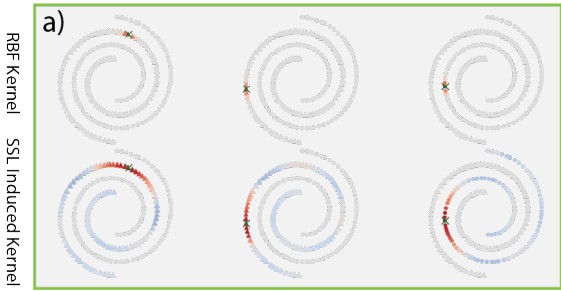 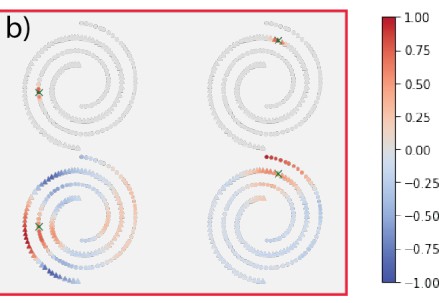

Figure 2: Comparison of the RBF kernel space (first row) and induced kernel space (second row). The induced kernel is computed based on Equation 4, and the graph Laplacian matrix is derived from the inner product neighborhood in the RBF kernel space, i.e., using the neighborhoods as data augmentation. a) We plot three randomly chosen points' kernel entries with respect to the other points on the manifolds. When the neighborhood augmentation range used to construct the Laplacian matrix is small enough, the SSL-induced kernel faithfully learns the topology of the entangled spiral manifolds. b) When the neighborhood augmentation range used to construct the Laplacian matrix is too large, it creates the "short-circuit" effect in the induced kernel space. Each subplot on the second row is normalized by its largest absolute value for better contrast.

$k(\cdot, \cdot)$ bounded in output between 0 and 1, the quantity $s_N(\boldsymbol{K})$ is minimized in binary classification when $k(\boldsymbol{x}, \boldsymbol{x}') = 1$ for $\boldsymbol{x}, \boldsymbol{x}'$ drawn from the same class and $k(\boldsymbol{x}, \boldsymbol{x}') = 0$ for $\boldsymbol{x}, \boldsymbol{x}'$ drawn from distinct classes. If the induced kernel works ideally – in the sense that it better correlates points within a class and decorrelates points otherwise – then the entries of the kernel matrix approach these optimal values. This intuition is also supported by the hypothesis that self-supervised and semi-supervised algorithms perform well by connecting the representations of points on a common data manifold (HaoChen et al., 2021; Belkin & Niyogi, 2004). To formalize this somewhat, consider such an ideal, yet fabricated, setting where the SSL induced kernel has complexity $s_N(\boldsymbol{K}^*)$ that does not grow with the dataset size.

**Proposition 3.3** (Ideal SSL outcome). *Given a supervised dataset of $N$ points for binary classification drawn from a distribution with $m_{-1}$ and $m_{+1}$ connected manifolds for classes with labels $-1$ and $+1$ respectively, if the induced kernel matrix of the dataset $\boldsymbol{K}^*$ successfully separates the manifolds such that $k^*(\boldsymbol{x}, \boldsymbol{x}') = 1$ if $\boldsymbol{x}, \boldsymbol{x}'$ are in the same manifold and $k^*(\boldsymbol{x}, \boldsymbol{x}') = 0$ otherwise, then $s_N(\boldsymbol{K}^*) = m_{-1} + m_{+1} = O(1)$.*

The simple proof of the above is in Appendix C. In short, we conjecture that SSL should be preferred in such settings where the relationships between datapoints are "strong" enough to connect similar points in a class on the same manifold. We analyze the quantity $s_N(\boldsymbol{K})$ in Appendix D.4 to add further empirical evidence behind this hypothesis.

## 4 EXPERIMENTS

In this section, we empirically investigate the performance and properties of the SSL kernel methods on a toy spiral dataset and portions of the MNIST and eMNIST datasets for hand-drawn digits and characters (Cohen et al., 2017). As with other works, we focus on small-data tasks where kernel methods can be performed efficiently without modifications needed for handling large datasets (Arora et al., 2019a; Fernández-Delgado et al., 2014). For simplicity and ease of analysis, we perform experiments here with respect to the RBF kernel. Additional experiments reinforcing these findings and also including analysis with neural tangent kernels can be found in Appendix D.

### 4.1 VISUALIZING THE INDUCED KERNEL ON THE SPIRAL DATASET

For an intuitive understanding, we provide a visualization in Figure 2 to show how the SSL-induced kernel disentangles manifolds in the representation space. In Figure 2, we study two entangled 1-D spiral manifolds in a 2D space with 200 training points uniformly distributed on the spiral manifolds. We use the non-contrastive SSL-induced kernel, following Equation (4), to demonstrate this result,

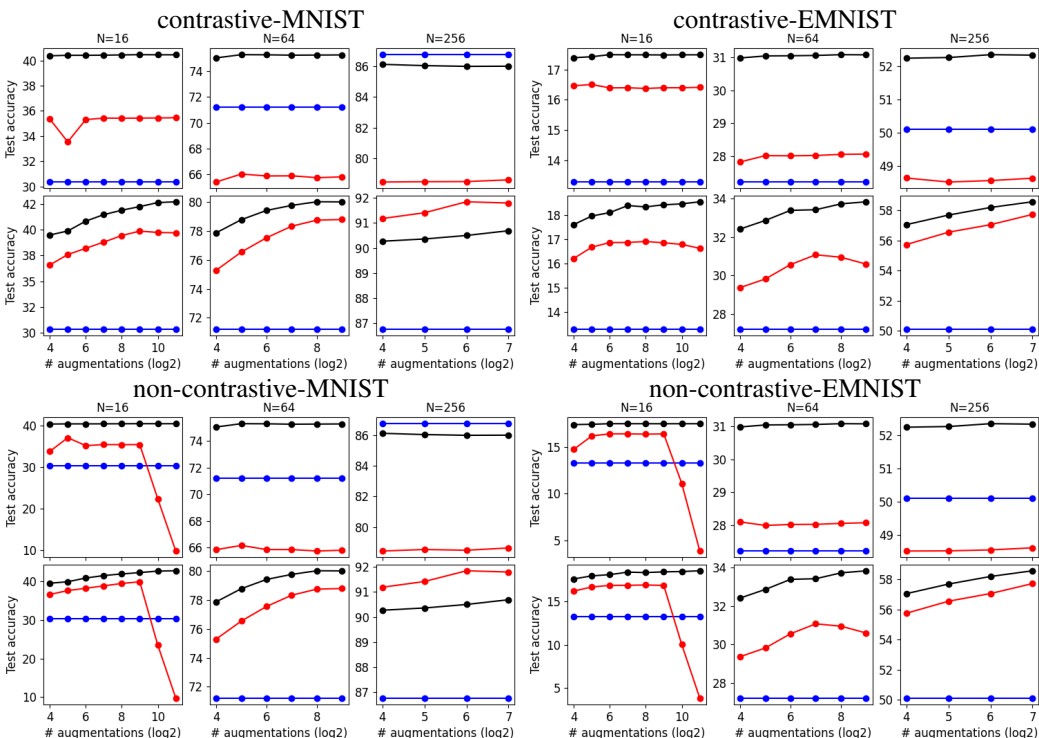

Figure 3: Depiction of MNIST and EMNIST full test set performances using the contrastive and non-contrastive kernels (**in red**) and benchmarked against the supervised case with labels on all samples (original + augmented) **in black** and with labels only on the original samples **in blue**, with the number of original samples given by $N \in \{16, 64, 256\}$ (each **column**) and the number of augmented samples (in log2) in **x-axis**. The **first row** corresponds to Gaussian blur data-augmentation (poorly aligned with the data distributions) and the **second row** corresponds to random rotations (-10,10), translations (-0.1,0.1) and scaling (0.9,1.1). We set the SVM $\ell_2$ regularization to $0.001$ and use the RBF kernel (NTK kernels in Appendix D.3). In all cases, the kernel representation dimensions are unconstrained. Two key observations emerge. First, whenever the augmentation is not aligned with the data distribution, the SSL kernels falls below the supervised case, especially as $N$ increases. Second, when the augmentation is aligned with the data distribution, both SSL kernels are able to get close and even outperform the supervised benchmark with augmented labels.

whereas a contrastive SSL-induced kernel is qualitatively similar and left to Appendix D.1. In the RBF kernel space shown in the first row of Figure 2, the value of the kernel is captured purely by distance. Next, to consider the SSL setting, we construct the graph Laplacian matrix $\boldsymbol{L}$ by connecting vertices between any training points with $k(x_1, x_2) > d$, i.e., $\boldsymbol{L}_{ij} = -1$ if $k(x_i, x_j) > d$ and $\boldsymbol{L}_{ij} = 0$ otherwise. The diagonal entries of $\boldsymbol{L}$ are equal to the degree of the vertices, respectively. This construction can be viewed as using the Euclidean neighborhoods of $x$ as the data augmentation. We choose $\beta = 0.4$, where other choices within a reasonable range lead to similar qualitative results. In the second row of Figure 2, we show the induced kernel between selected points (marked with an x) and other training points in the SSL-induced kernel space. When $d$ is chosen properly, as observed in the second row of Figure 2(a), the SSL-induced kernel faithfully captures the topology of manifolds. However, the augmentation has to be carefully chosen as Figure 2(b) shows that when $d$ is too large, the two manifolds become mixed in the representation space.

## 4.2 CLASSIFICATION EXPERIMENTS

We explore in Figure 3 the supervised classification setting of MNIST and EMNIST which consist of $28 \times 28$ grayscale images. (E)MNIST provide a strong baseline to evaluate kernel methods due to the absence of background in the images making kernels such as RBF more aligned to measure input similarities. In this setting, we explore two different data-augmentation (DA) policies, one aligned

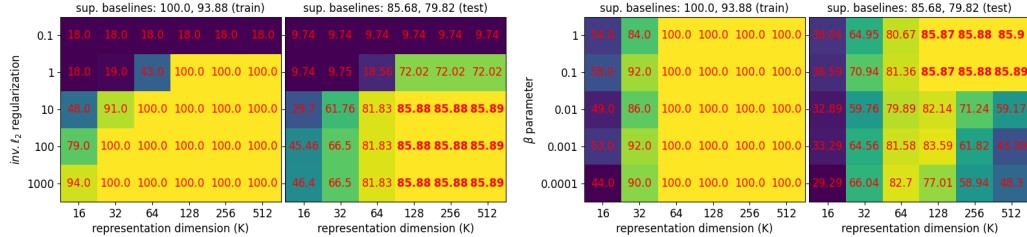

Figure 4: MNIST classification task with $100$ original training samples and $100$ augmentations per sample on the train and test set (full, $10,000$ samples) with baselines in the titles given by supervised SVM using labels for all samples or original ($100$) samples only. We provide on the **left** the contrastive kernel performances when ablating over the (inverse) of the SVM's $\ell_2$ regularizer **y-axis** and representation dimension ($K$) in the **x-axis**. Similarly, we provide on the **right** the non-contrastive kernel performances when ablating over $\beta$ on the **y-axis** and over the representation dimension ($K$) on the **x-axis**. We observe, as expected, that reducing $K$ prevents overfitting and should be preferred over the $\ell_2$ regularizer, and that $\beta$ acts jointly with the representation dimension i.e. one only needs to tune one of the two.

with the data distribution (rotation+translation+scaling) and one largely misaligned with the data distribution (aggressive Gaussian blur). Because our goal is to understand how much DA impacts the SSL kernel compared to a fully supervised benchmark, we consider two (supervised) benchmarks: one that employs the labels of the sampled training set and all the augmented samples and one that only employs the sampled training set and no augmented samples. We explore a small training set size going from $N = 16$ to $N = 256$ and for each case we produce a number of augmented samples for each datapoint so that the total number of samples does not exceed $50,000$ which is a standard threshold for kernel methods. We observed in Figure 3 that the SSL kernel is able to match and even outperform the fully supervised case when employing the correct data-augmentation, while with the incorrect data-augmentation, the performance is not even able to match the supervised case that did not see the augmented samples. To better understand the impact of different hyperparameters onto the two kernels, we also study in Figure 4 the MNIST test set performances when varying the representation dimension $K$, the SVM's $\ell_2$ regularization parameter, and the non-contrastive kernel's $\beta$ parameter. We observe that although $\beta$ is an additional hyper-parameter to tune, its tuning plays a similar role to $K$, the representation dimension. Hence, in practice, the design of the non-contrastive model can be controlled with a single parameter as with the contrastive setting. We also interestingly observe that $K$ is a more preferable parameter to tune to prevent over-fitting as opposed to SVM's $\ell_2$ regularizer.

## 5 DISCUSSION

Our work explores the properties of SSL algorithms when trained via kernel methods. Connections between kernel methods and neural networks have gained significant interest in the supervised learning setting (Neal, 1996; Lee et al., 2017) for their potential insights into the training of deep networks. As we show in this study, such insights into the training properties of SSL algorithms can similarly be garnered from an analysis of SSL algorithms in the kernel regime. Our theoretical and empirical analysis, for example, highlights the importance of the choice of augmentations and encoded relationships between data points on downstream performance. Looking forward, we believe that interrelations between this kernel regime and the actual deep networks trained in practice can be strengthened particularly by analyzing the neural tangent kernel. In line with similar analysis in the supervised setting (Yang et al., 2022; Seleznova & Kutyniok, 2022; Lee et al., 2020), neural tangent kernels and their corresponding induced kernels in the SSL setting may shine light on some of the theoretical properties of the finite width networks used in practice.

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

# A  RELATED WORKS AND TOPICS

## A.1  JOINT EMBEDDING SSL ALGORITHMS IN PRACTICE

Joint embedding approaches form representations by comparing representations of jointly chosen inputs that have known relations. These methods also can hold out portions of the data, but this is not necessarily a requirement. Any joint-embedding SSL algorithm requires a properly chosen loss function and access to a set of observations and known pairwise positive relation between those observations. Methods are denoted as non-contrastive if the loss function is only a function of pairs that are related (Grill et al., 2020; Chen & He, 2021; Zbontar et al., 2021). One common method using the VICReg loss (Bardes et al., 2021), for example, takes the form

$$\mathcal{L}_{\text{vic}} = \alpha \sum_{k=1}^{K} \max\left(0, 1 - \sqrt{\text{Cov}(\boldsymbol{Z})_{k,k}}\right) + \beta \sum_{j=1, j \neq k}^{K} \text{Cov}(\boldsymbol{Z})_{k,j}^2 + \frac{\gamma}{N} \sum_{i=1}^{N} \sum_{j=1}^{N} (\boldsymbol{A})_{i,j} \|\boldsymbol{Z}_{i,.} - \boldsymbol{Z}_{j,.}\|_2^2.$$

(14)

We adapt the above for the non-contrastive loss we study in our work.

Contrastive methods also penalize similarities of representations that are not related. Popular algorithms include SimCLR (Chen et al., 2020), SwAV (Caron et al., 2020), NNCLR (Dwibedi et al., 2021), contrastive predictive coding (Oord et al., 2018), and many others. The spectral contrastive loss, for example takes the form (HaoChen et al., 2021):

$$\mathcal{L}_{sc} = -2\mathbb{E}_{x,x^+}\left[f(x)^{\mathsf{T}} f(x^+)\right] + \mathbb{E}_{x,x^-}\left[\left(f(x)^{\mathsf{T}} f(x^-)\right)^2\right],$$

(15)

where $x, x^+$ are positive pairs and $x, x^-$ are negative pairs. We consider a variant of the spectral contrastive loss in our work.

## A.2  THEORETICAL STUDIES OF SSL

In tandem with the success of SSL in deep learning, a host of theoretical tools have been developed to help understand how SSL algorithms learn (Arora et al., 2019b; Balestriero & LeCun, 2022; HaoChen et al., 2022; Lee et al., 2021). Findings are often connected to the underlying graph connecting the data distribution (Wei et al., 2020; HaoChen et al., 2021) or the choice of augmentation (Wen & Li, 2021). Wang & Isola (2020) study the theoretical properties of good SSL algorithms showing that alignment and uniformity are two important criteria for successfully building representations. We also note that prior work on SSL has analyzed the importance of the choice of augmentation in building positive pairs (Zheng et al., 2021; Von Kügelgen et al., 2021). Building representations from known relationships between datapoints is also studied in spectral graph theory (Chung, 1997). We employ findings from this body of literature to provide intuition behind the properties of the algorithms discussed here.

## A.3  KERNEL METHODS

Kernel methods have been connected to properties of graphs and manifolds in a long line of work. Smola & Kondor (2003); Vishwanathan et al. (2010), among others, introduce a family of kernels on graphs which offer nice regularization properties for their corresponding RKHS. Similar methods have been proposed for semi-supervised learning via graph-based methods (Zhu, 2005; Hofmann et al., 2008) and manifold regularization (Belkin et al., 2006). These methods generally introduce terms into a loss function which regularize the solution depending on the geometric structure of the data. Properties of the features have also been tied to results in spectral graph theory (Zhao & Liu, 2007; 2012; Dong et al., 2012), where the top features generally correspond to the top eigenvalues of the graph Laplacian. More recently, kernel methods have been employed to more efficiently regularize data based on the Laplacian and improve semi-supervised learning algorithms (Cabannes et al., 2021; Cabannes, 2022).

As stated in the main text, perhaps the closest lines of work are related to kernel and metric learning (Bellet et al., 2013; Yang & Jin, 2006) In kernel learning, prior works have proposed constructing a kernel via a learning procedure; e.g., via convex combinations of kernels (Cortes et al., 2010), kernel

alignment (Cristianini et al., 2001), and unsupervised kernel learning to match local data geometry (Zhuang et al., 2011). Prior work in distance metric learning using kernel methods aim to produce representations of data in unsupervised or semi-supervised settings by taking advantage of links between data points. For example, (Baghshah & Shouraki, 2010; Hoi et al., 2007) learn to construct a kernel based on optimizing distances between points embedded in Hilbert space according to a similarity and dissimilarity matrix. Yeung & Chang (2007) perform kernel distance metric learning in a semi-supervised setting where pairwise relations between data and labels are provided. Xia et al. (2013) propose an online procedure to learn a kernel which maps similar points closer to each other than dissimilar points. Many of these works also use semi-definite programs to perform optimization to find the optimal kernel.

# B  DEFERRED PROOFS

## B.1  REPRESENTER THEOREM

**Theorem B.1** (Representer theorem for self-supervised tasks; adapted from Schölkopf et al. (2001))**.**
*Let $\boldsymbol{x}_i \in \mathcal{X}$ for $i \in [N]$ be elements of a dataset of size $N$, $k : \mathcal{X} \times \mathcal{X} \to \mathbb{R}$ be a kernel function with corresponding map $\Phi : \mathcal{X} \to \mathcal{H}$ into the RKHS $\mathcal{H}$, and $r : [0, \infty) \to \mathbb{R}$ be any strictly increasing function. Let $\boldsymbol{W} : \mathcal{H} \to \mathbb{R}^K$ be a linear function mapping inputs to their corresponding representations. Given a regularized loss function of the form*

$$\mathcal{R}\left(\boldsymbol{W}\Phi(\boldsymbol{x}_1), \ldots, \boldsymbol{W}\Phi(\boldsymbol{x}_N)\right) + r\left(\|\boldsymbol{W}\|_{\mathcal{H}}\right), \tag{16}$$

*where $\mathcal{R}(\boldsymbol{W}\Phi(\boldsymbol{x}_1), \ldots, \boldsymbol{W}\Phi(\boldsymbol{x}_N))$ is an error function that depends on the representations of the dataset, the minimizer $\boldsymbol{W}^*$ of this loss function will be in the span of the training points $\{\Phi(\boldsymbol{x}_i),\ i \in \{1, \ldots, N\}\}$, i.e. for any $\boldsymbol{\phi} \in \mathcal{H}$:*

$$\boldsymbol{W}^*\boldsymbol{\phi} = \boldsymbol{0}\ \ if\ \ \langle \boldsymbol{\phi}, \Phi(\boldsymbol{x}_i)\rangle_{\mathcal{H}} = 0\ \ \forall i \in [N]. \tag{17}$$

*Proof.* Decompose $\boldsymbol{W} = \boldsymbol{W}_{\|} + \boldsymbol{W}_{\perp}$, where $\boldsymbol{W}_{\perp}\Phi(\boldsymbol{x}_i) = \boldsymbol{0}$ for all $i \in [N]$. For a loss function $\mathcal{L}$ of the form listed above, we have

$$
\begin{aligned}
\mathcal{L}(\boldsymbol{W}_{\|} + \boldsymbol{W}_{\perp}) &= \mathcal{R}\left((\boldsymbol{W}_{\|} + \boldsymbol{W}_{\perp})\Phi(\boldsymbol{x}_1), \ldots, (\boldsymbol{W}_{\|} + \boldsymbol{W}_{\perp})\Phi(\boldsymbol{x}_N)\right) + r\left(\|\boldsymbol{W}_{\|} + \boldsymbol{W}_{\perp}\|_{\mathcal{H}}\right) \\
&= \mathcal{R}\left(\boldsymbol{W}_{\|}\Phi(\boldsymbol{x}_1), \ldots, \boldsymbol{W}_{\|}\Phi(\boldsymbol{x}_N)\right) + r\left(\|\boldsymbol{W}_{\|} + \boldsymbol{W}_{\perp}\|_{\mathcal{H}}\right).
\end{aligned} \tag{18}
$$

Where in the terms in the sum, we used the property that $\boldsymbol{W}_{\perp}$ is not in the span of the data. For the regularizer term, we note that

$$
\begin{aligned}
r(\|\boldsymbol{W}_{\|} + \boldsymbol{W}_{\perp}\|_{\mathcal{H}}) &= r\left(\sqrt{\|\boldsymbol{W}_{\|}\|_{\mathcal{H}}^2 + \|\boldsymbol{W}_{\perp}\|_{\mathcal{H}}^2}\right) \\
&\geq r(\|\boldsymbol{W}_{\|}\|_{\mathcal{H}}).
\end{aligned} \tag{19}
$$

Therefore, strictly enforcing $\boldsymbol{W}^{\perp} = 0$ minimizes the regularizer while leaving the rest of the cost function unchanged. $\square$

As a consequence of the above, all optimal solutions must have support over the span of the data. This directly results in the statement shown in Proposition 3.1.

*Remark* B.2. The representer theorem renders the problem of optimization tractable over a finite dataset for a given SSL loss function (Hofmann et al., 2008; Soman et al., 2009). In certain cases, as we show in the main text, such optimization have nice closed form solutions. Nevertheless, approximate methods of optimization can be used to find closed form solutions for loss functions where closed form solutions do not exist or are not expected to exist. Many optimizers exist to perform this optimization including potentially approximations that allow for incorporation of larger datasets (Hofmann et al., 2008; Deisenroth & Ng, 2015; Jain et al., 2012; Menon, 2009; Rasmussen, 2003).

## B.2 Closed form non-contrastive loss

Throughout this section, for simplicity, we define $M := I - \frac{1}{N}\mathbf{1}\mathbf{1}^\mathsf{T}$. With slight abuse of notation, we also denote $X \in \mathbb{R}^{N \times D}$ as a matrix whose $i$-th row contains the features of $\Phi(\boldsymbol{x}_i)$.

$$X = \begin{bmatrix} - & \Phi(\boldsymbol{x}_1)^\mathsf{T} & - \\ - & \Phi(\boldsymbol{x}_2)^\mathsf{T} & - \\ & \vdots & \\ - & \Phi(\boldsymbol{x}_N)^\mathsf{T} & - \end{bmatrix} \tag{20}$$

Note that if the RKHS is infinite dimensional, one can apply the general form of the solution as shown in Proposition 3.1 to reframe the problem into the finite dimensional setting below. As a reminder, we aim to minimize the VICreg cost function:

$$C^* = \min_{\boldsymbol{W} \in \mathbb{R}^{K \times D}} \|WX^\mathsf{T}MXW^\mathsf{T} - I\|_F^2 + \beta \operatorname{Tr}\left[WX^\mathsf{T}LXW^\mathsf{T}\right]. \tag{21}$$

By applying the definition of the Frobenius norm and from some algebra, we obtain:

$$\begin{aligned} C(W) &= \|WX^\mathsf{T}MXW^\mathsf{T} - I\|_F^2 + \beta \operatorname{Tr}\left[WX^\mathsf{T}LXW^\mathsf{T}\right] \\ &= \|WX^\mathsf{T}MXW^\mathsf{T} - I\|_F^2 + \operatorname{Tr}\left[WX^\mathsf{T}\left(\beta L + 2M - 2M\right)XW^\mathsf{T}\right] \\ &= K + \|WX^\mathsf{T}MXW^\mathsf{T}\|_F^2 - 2\operatorname{Tr}\left[WX^\mathsf{T}MXW^\mathsf{T}\right] \\ &\quad + \operatorname{Tr}\left[WX^\mathsf{T}\left(\beta L + 2M - 2M\right)XW^\mathsf{T}\right] \\ &= K + \|WX^\mathsf{T}MXW^\mathsf{T}\|_F^2 - \operatorname{Tr}\left[WX^\mathsf{T}\left(2M - \beta L\right)XW^\mathsf{T}\right] \\ &= K + \|XW^\mathsf{T}WX^\mathsf{T}M\|_F^2 - \operatorname{Tr}\left[XW^\mathsf{T}WX^\mathsf{T}M\left(2M - \beta L\right)\right]. \end{aligned} \tag{22}$$

The optimum of the above formulation has the same optimum as the following optimization problem defined as $C'(W)$:

$$C'(W) = \|XW^\mathsf{T}WX^\mathsf{T}M - \left(M - \beta L/2\right)\|_F^2. \tag{23}$$

Since $M$ is a projector and $M(2M - \beta L) = 2M - \beta L$ (since the all-ones vector is in the kernel of $L$), then we can solve the above by employing the Eckart-Young theorem and matching the $K$-dimensional eigenspace of $XW^\mathsf{T}WX^\mathsf{T}M$ with that of the top $K$ eigenspace of $M - \beta L/2$.

One must be careful in choosing this optimum as $WX^\mathsf{T}MXW^\mathsf{T}$ can only take positive eigenvalues. Therefore, this is achieved by choosing the optimal $W^*$ to project the data onto this eigenspace as

$$W^* = \left[X^{(s)^+}C_{:,\leq K}\left(I - D\right)_{\leq K,\leq K}^{1/2}\right]^\mathsf{T}, \tag{24}$$

where we set the eigendecomposition of $\frac{1}{N}\mathbf{1}\mathbf{1}^\mathsf{T} + \frac{\beta}{2}L$ as $CDC^\mathsf{T} = \frac{1}{N}\mathbf{1}\mathbf{1}^\mathsf{T} + \frac{\beta}{2}L$ and $C_{:,\leq K}$ is the matrix consisting of the first $K$ rows of $C$. Similarly, $(I - D)_{\leq K,\leq K}^{1/2}$ denotes the top left $K \times K$ matrix of $(I - D)$. Also in the above, $X^{(s)^+}$ denotes the pseudo-inverse of $X$. If the diagonal matrix $I - D$ contains negative entries, which can only happen when $\beta$ is set to a large value, then the values of $(I - D)^{1/2}$ for those entries is undefined. Here, the optimum choice is to set those entries to equal zero. In practice, this can be avoided by setting $\beta$ to be no larger two times than the degree of the graph.

Note, that since $W$ only appears in the cost function in the form $W^\mathsf{T}W$, the solution above is only unique up to an orthogonal transformation. Furthermore, the rank of $XW^\mathsf{T}WX^\mathsf{T}M$ is at most $N - 1$ so a better optimum cannot be achieved by increasing the output dimension of the linear transformation beyond $N$. To see that this produces the optimal induced kernel, we simply plug in the optimal $W^*$:

$$\begin{aligned} k^*(\boldsymbol{x}, \boldsymbol{x}') &:= (W^*\boldsymbol{x})^\mathsf{T}(W^*\boldsymbol{x}') \\ &= \boldsymbol{x}^\mathsf{T}X^{(s)^+}C_{:,\leq K}\left(I - D\right)_{\leq K,\leq K}^{1/2}\left(I - D\right)_{\leq K,\leq K}^{1/2}C_{:,\leq K}^\mathsf{T}\left(X^{(s)^+}\right)^\mathsf{T}\boldsymbol{x}' \\ &= \boldsymbol{k}_{\boldsymbol{x},s}K_{s,s}^{-1}C_{:,\leq K}\left(I - D\right)_{\leq K,\leq K}C_{:,\leq K}^\mathsf{T}K_{s,s}^{-1}\boldsymbol{k}_{s,\boldsymbol{x}'}. \end{aligned} \tag{25}$$

Now, it needs to be shown that $\boldsymbol{W}^*$ is the unique norm optimizer of the optimization problem. To show this, we analyze the following semi-definite program which is equivalent since the cost function is over positive semi-definite matrices of the form $\boldsymbol{W}^\intercal \boldsymbol{W}$:

$$\min_{\boldsymbol{B} \in \mathbb{R}^{D \times D}} \operatorname{Tr}(\boldsymbol{B})$$
$$\text{s.t. } \boldsymbol{XBX}^\intercal \boldsymbol{M} = \boldsymbol{C}_{:,\leq K} (\boldsymbol{I} - \boldsymbol{D})^{1/2}_{\leq K, \leq K} \boldsymbol{C}^\intercal_{:,\leq K} := \boldsymbol{P}_K \tag{26}$$
$$\boldsymbol{B} \succeq 0$$

To lighten notation, we denote $\boldsymbol{P}_K = \boldsymbol{C}_{:,\leq K} (\boldsymbol{I} - \boldsymbol{D})^{1/2}_{\leq K, \leq K} \boldsymbol{C}^\intercal_{:,\leq K}$ and simply use $\boldsymbol{X}$ for $\boldsymbol{X}$. The above can easily be derived by setting $\boldsymbol{B} = \boldsymbol{W}^\intercal \boldsymbol{W}$ in Equation (23). This has corresponding dual

$$\max_{\boldsymbol{Y} \in \mathbb{R}^{N \times N}} \operatorname{Tr}(\boldsymbol{Y}^\intercal \boldsymbol{P}_K)$$
$$\text{s.t. } \boldsymbol{I} - \boldsymbol{X}^\intercal \boldsymbol{Y} \boldsymbol{M} \boldsymbol{X} \succeq 0 \tag{27}$$

The optimal primal can be obtained from $\boldsymbol{W}^*$ by $\boldsymbol{B}^* = (\boldsymbol{W}^*)^\intercal \boldsymbol{W}^* = \boldsymbol{X}^+ \boldsymbol{P}_K (\boldsymbol{X}^+)^\intercal$. The optimal dual can be similarly calculated and is equal to $\boldsymbol{Y}^* = (\boldsymbol{X}^+)^\intercal \boldsymbol{X}^+$. A straightforward calculation shows that the optimum value of the primal and dual formulation are equal for the given solutions. We now check whether the chosen solutions of the primal and dual satisfy the KKT optimality conditions (Boyd et al., 2004):

$$\text{Primal feasibility: } \boldsymbol{XB}^* \boldsymbol{X}^\intercal \boldsymbol{M} = \boldsymbol{P}_K, \ \ \boldsymbol{B}^* \succeq 0$$
$$\text{Dual feasibility: } \boldsymbol{I} - \boldsymbol{X}^\intercal \boldsymbol{Y}^* \boldsymbol{M} \boldsymbol{X} \succeq 0 \tag{28}$$
$$\text{Complementary slackness: } (\boldsymbol{I} - \boldsymbol{X}^\intercal \boldsymbol{Y}^* \boldsymbol{M} \boldsymbol{X}) \boldsymbol{B}^* = 0.$$

The primal feasibility and dual feasibility criteria are straightforward to check. For complementary slackness, we note that

$$\begin{aligned} (\boldsymbol{I} - \boldsymbol{X}^\intercal \boldsymbol{Y}^* \boldsymbol{M} \boldsymbol{X}) \boldsymbol{B}^* &= \boldsymbol{X}^+ \boldsymbol{L}_K (\boldsymbol{X}^+)^\intercal - \boldsymbol{X}^\intercal (\boldsymbol{X}^+)^\intercal \boldsymbol{X}^+ \boldsymbol{M} \boldsymbol{X} \boldsymbol{X}^+ \boldsymbol{L}_K (\boldsymbol{X}^+)^\intercal \\ &= \boldsymbol{X}^+ \boldsymbol{L}_K (\boldsymbol{X}^+)^\intercal - \boldsymbol{X}^+ \boldsymbol{M} \boldsymbol{L}_K (\boldsymbol{X}^+)^\intercal \\ &= \boldsymbol{X}^+ \boldsymbol{L}_K (\boldsymbol{X}^+)^\intercal - \boldsymbol{X}^+ \boldsymbol{L}_K (\boldsymbol{X}^+)^\intercal \\ &= 0. \end{aligned} \tag{29}$$

In the above we used the fact that $\boldsymbol{M} \boldsymbol{L}_K = \boldsymbol{L}_K$ since $\boldsymbol{M}$ is a projector and $\boldsymbol{L}_K$ is unchanged by that projection. This completes the proof of the optimality.

### B.3 CONTRASTIVE LOSS

For the contrastive loss, we follow a similar approach as above to find the minimum norm solution that obtains the optimal representation. Note, that the loss function contains the term $\|\boldsymbol{X} \boldsymbol{W}^\intercal \boldsymbol{W} \boldsymbol{X}^\intercal - (\boldsymbol{I} + \boldsymbol{A})\|^2_F$. Since $\boldsymbol{X} \boldsymbol{W}^\intercal \boldsymbol{W} \boldsymbol{X}^\intercal$ is positive semi-definite, then this is optimized when $\boldsymbol{X} \boldsymbol{W}^\intercal \boldsymbol{W} \boldsymbol{X}^\intercal$ matches the positive eigenspace of $(\boldsymbol{I} + \boldsymbol{A})$ defined as $(\boldsymbol{I} + \boldsymbol{A})_+$. Enumerating the eigenvalues of $\boldsymbol{A}$ as $\boldsymbol{v}_i$ with corresponding eigenvalues $e_i$, then $(\boldsymbol{I} + \boldsymbol{A})_+ = \sum_{i:e_i \geq -1} (e_i + 1) \boldsymbol{v}_i \boldsymbol{v}_i^\dagger$. More generally, if the dimension of the representation is restricted such that $K < N$, then we abuse notation and define

$$(\boldsymbol{I} + \boldsymbol{A})_+ = \sum_{i=1}^{K} \max(e_i + 1, 0) \boldsymbol{v}_i \boldsymbol{v}_i^\dagger, \tag{30}$$

where $e_i$ are sorted in descending order.

To find the minimum RKHS norm solution, we have to solve a similar SDP to Equation (26):

$$\min_{\boldsymbol{B} \in \mathbb{R}^{D \times D}} \operatorname{Tr}(\boldsymbol{B})$$
$$\text{s.t. } \boldsymbol{XBX}^\intercal = (\boldsymbol{I} + \boldsymbol{A})_+ \tag{31}$$
$$\boldsymbol{B} \succeq 0$$

This has corresponding dual

$$
\max_{\boldsymbol{Y} \in \mathbb{R}^{N \times N}} \operatorname{Tr} \left( \boldsymbol{Y}^{\mathsf{T}} (\boldsymbol{I} + \boldsymbol{A})_+ \right)
$$
$$
\text{s.t. } \boldsymbol{I} - \boldsymbol{X}^{\mathsf{T}} \boldsymbol{Y} \boldsymbol{X} \succeq 0 \tag{32}
$$

The optimal primal is $\boldsymbol{B}^* = \boldsymbol{X}^+ (\boldsymbol{I} + \boldsymbol{A})_+ \left( \boldsymbol{X}^+ \right)^{\mathsf{T}}$. The optimal dual is equal to $\boldsymbol{Y}^* = \left( \boldsymbol{X}^+ \right)^{\mathsf{T}} \boldsymbol{X}^+$. Directly plugging these in shows that the optimum value of the primal and dual formulation are equal for the given solutions. As before, we now check whether the chosen solutions of the primal and dual satisfy the KKT optimality conditions (Boyd et al., 2004):

$$
\text{Primal feasibility: } \boldsymbol{X} \boldsymbol{B}^* \boldsymbol{X}^{\mathsf{T}} = (\boldsymbol{I} + \boldsymbol{A})_+ , \;\; \boldsymbol{B}^* \succeq 0
$$
$$
\text{Dual feasibility: } \boldsymbol{I} - \boldsymbol{X}^{\mathsf{T}} \boldsymbol{Y}^* \boldsymbol{X} \succeq 0 \tag{33}
$$
$$
\text{Complementary slackness: } \left( \boldsymbol{I} - \boldsymbol{X}^{\mathsf{T}} \boldsymbol{Y}^* \boldsymbol{X} \right) \boldsymbol{B}^* = 0.
$$

The primal feasibility and dual feasibility criteria are straightforward to check. For complementary slackness, we note that

$$
\begin{aligned}
\left( \boldsymbol{I} - \boldsymbol{X}^{\mathsf{T}} \boldsymbol{Y}^* \boldsymbol{X} \right) \boldsymbol{B}^* &= \boldsymbol{X}^+ (\boldsymbol{I} + \boldsymbol{A})_+ \left( \boldsymbol{X}^+ \right)^{\mathsf{T}} - \boldsymbol{X}^{\mathsf{T}} \left( \boldsymbol{X}^+ \right)^{\mathsf{T}} \boldsymbol{X}^+ \boldsymbol{X} \boldsymbol{X}^+ (\boldsymbol{I} + \boldsymbol{A})_+ \left( \boldsymbol{X}^+ \right)^{\mathsf{T}} \\
&= \boldsymbol{X}^+ (\boldsymbol{I} + \boldsymbol{A})_+ \left( \boldsymbol{X}^+ \right)^{\mathsf{T}} - \boldsymbol{X}^+ (\boldsymbol{I} + \boldsymbol{A})_+ \left( \boldsymbol{X}^+ \right)^{\mathsf{T}} \\
&= 0.
\end{aligned} \tag{34}
$$

In the above, we used the fact that $\boldsymbol{X}^+ \boldsymbol{X}$ is a projection onto the row space of $\boldsymbol{X}$. This completes the proof of the optimality.

### B.4 Optimization via semi-definite program

In general scenarios, Proposition 3.1 gives a prescription for calculating the optimal induced kernel for more complicated optimization tasks since we note that the optimal induced kernel $k^*(\boldsymbol{x}, \boldsymbol{x}')$ must be of the form below:

$$
k^*(\boldsymbol{x}, \boldsymbol{x}') = \boldsymbol{k}_{\boldsymbol{x},s} \boldsymbol{B} \boldsymbol{k}_{\boldsymbol{x}',s}^{\mathsf{T}}, \tag{35}
$$

where $\boldsymbol{k}_{\boldsymbol{x},s}$ is a row vector whose entry $i$ equals the kernel $k(\boldsymbol{x}, \boldsymbol{x}_i)$ between $\boldsymbol{x}$ and the $i$-th datapoint and $\boldsymbol{B} \in \mathbb{R}^{N \times N}$ is a positive semi-definite matrix.

For example, such a scenario arises when one wants to apply the loss function across $n_{\text{batches}}$ batches of data. To frame this as an optimization statement, assume we have $N$ datapoints split into batches of size $b$ each. We denote the $i$-th datapoint within batch $k$ as $\boldsymbol{x}_i^{(k)}$. As before, $\boldsymbol{x}_i$ denotes the $i$-th datapoint across the whole dataset. We define the following variables:

- $\boldsymbol{K}_{s,s} \in \mathbb{R}^{N \times N}$ (kernel matrix over complete dataset including all batches) where $[\boldsymbol{K}_{s,s}]_{i,j} = k(\boldsymbol{x}_i, \boldsymbol{x}_j)$

- $\boldsymbol{K}_{s,s_k} \in \mathbb{R}^{N \times b}$ (kernel matrix between complete dataset and dataset of batch k) where $[\boldsymbol{K}_{s,s_k}]_{i,j} = k(\boldsymbol{x}_i, \boldsymbol{x}_j^{(k)})$; similarly, $\boldsymbol{K}_{s_k,s} \in \mathbb{R}^{b \times N}$ is simply the transpose of $\boldsymbol{K}_{s,s_k}$

- $\boldsymbol{A}^{(k)}$ is the adjacency matrix for inputs in batch $k$ with corresponding graph Laplacian $\boldsymbol{L}^{(k)}$

In what follows, we denote the representation dimension as $K$.

**Non-contrastive loss function** In the non-contrastive setting, we consider a regularized version of the batched loss of Equation (3). Applying the reduction of the loss function in Equation (23), we consider the following loss function where we want to find the minimizer $\boldsymbol{B}$:

$$
\mathcal{L} = \sum_{j=1}^{n_{\text{batches}}} \left\| \boldsymbol{K}_{s_j,s} \boldsymbol{B} \boldsymbol{K}_{s,s_j} \left( \boldsymbol{I} - \frac{1}{b} \mathbf{1} \mathbf{1}^{\mathsf{T}} \right) - \left( \boldsymbol{I} - \frac{1}{b} \mathbf{1} \mathbf{1}^{\mathsf{T}} - \beta \boldsymbol{L}^{(j)} / 2 \right) \right\|_F^2 + \alpha \operatorname{Tr}(\boldsymbol{B} \boldsymbol{K}_{s,s}), \tag{36}
$$

where $\alpha \in \mathbb{R}^+$ is a hyperparameter. The term $\operatorname{Tr}(\boldsymbol{B} \boldsymbol{K}_{s,s})$ regularizes for the RKHS norm of the resulting solution given by $\boldsymbol{B}$. For simplicity, we denote as before $\boldsymbol{M} = \boldsymbol{I} - \frac{1}{b} \mathbf{1} \mathbf{1}^{\mathsf{T}}$. Taking the limit

$\alpha \to 0$ and enforcing a representation of dimension $K$, the loss function above is minimized when we obtain the optimal representation, i.e. we must have that

$$K_{s_j,s}BK_{s,s_j}M = \left(M - \beta L^{(j)}/2\right)_K,\tag{37}$$

where $\left(M - \beta L^{(j)}/2\right)_K$ denotes the projection of $M - \beta L^{(j)}/2$ onto the eigenspace of the top $K$ positive singular values (this is the optimal representation as shown earlier).

Therefore, we can find the optimal induced kernel by solving the optimization problem below:

$$\min_{B \in \mathbb{R}^{N \times N}} \text{Tr}(BK_{s,s})$$
$$\text{s.t. } K_{s_i,s}BK_{s,s_i}M = \left(M - \beta L^{(j)}/2\right)_K \quad \forall i \in \{1, 2, \dots, n_{\text{batches}}\}\tag{38}$$
$$B \succeq 0, \text{ rank}(B) = K,$$

Relaxing the constraint $\text{rank}(B) = K$ forms an SDP which can be solved efficiently using existing SDP solvers (ApS, 2019; Sturm, 1999).

**Contrastive loss**    As shown in Section 3.1, the contrastive loss function takes the form

$$\mathcal{L} = \sum_{j=1}^{n_{\text{batches}}} \left\| K_{s_j,s}BK_{s,s_j} - \left(I + A^{(j)}\right) \right\|_F^2 + \alpha \text{ Tr}(BK_s, s)\tag{39}$$

, where $\alpha \in \mathbb{R}^+$ is a weighting term for the regularizer. In this setting, the optimal representation of dimension $K$ is equal to

$$K_{s_j,s}BK_{s,s_j} - \left(I + A^{(j)}\right)_K,\tag{40}$$

where $\left(I + A^{(j)}\right)_K$ denotes the projection of $I + A^{(j)}$ onto the eigenspace of the top $K$ positive singular values (this is the optimal representation as shown earlier). Taking the limit of $\alpha \to 0$, have a similar optimization problem:

$$\min_{B \in \mathbb{R}^{N \times N}} \text{Tr}(BK_{s,s})$$
$$\text{s.t. } K_{s_i,s}BK_{s,s_i} = \left(I + A^{(i)}\right)_K \quad \forall i \in \{1, 2, \dots, n_{\text{batches}}\}\tag{41}$$
$$B \succeq 0, \text{ rank}(B) = K.$$

As before, relaxing the rank constraint results in an SDP.

### B.5   Proof of kernel closeness

Our goal is to prove Proposition 3.2 copied below.

**Proposition 3.2.** *Given kernel function $k(\cdot, \cdot)$ with corresponding map $\Phi(\cdot)$ into the RKHS $\mathcal{H}$, let $\{x_1, x_2, \dots x_N\}$ be an SSL dataset normalized such that $k(x_i, x_i) = 1$ and formed by pairwise augmentations (i.e., every element has exactly one neighbor in $A$) with kernel matrix $K_{s,s}$. Given two points $x$ and $x'$, if there exists two points in the SSL dataset indexed by $i$ and $j$ which are related by an augmentation ($A_{ij}=1$) and $\|\Phi(x) - \Phi(x_i)\|_{\mathcal{H}} \leq \frac{\Delta}{5\|K_{s,s}^{-1}\|\sqrt{N}}$ and $\|\Phi(x') - \Phi(x_j)\|_{\mathcal{H}} \leq \frac{\Delta}{5\|K_{s,s}^{-1}\|\sqrt{N}}$, then the induced kernel for the contrastive loss is at least $k^*(x, x') \geq 1 - \Delta$.*

Note, the adjacency matrix $A$ for pairwise augmentations takes the form below assuming augmented samples are placed next to each other in order.

$$A = \begin{bmatrix} 0 & 1 & & & & & \\ 1 & 0 & & & & & \\ & & 0 & 1 & & & \\ & & 1 & 0 & & & \\ & & & & \ddots & & \\ & & & & & 0 & 1 \\ & & & & & 1 & 0 \end{bmatrix}.\tag{42}$$

Before proceeding, we prove a helper lemma that shows that $\|k_{s,x_a} - k_{s,x_b}\|$ is small if $\|\Phi(x_a) - \Phi(x_b)\|$ is also relatively small with a factor of dependence on the dataset size.

**Lemma B.3.** *Given kernel function $k(\cdot, \cdot)$ with map $\Phi(\cdot)$ into RKHS $\mathcal{H}$ and an SSL dataset $\{\boldsymbol{x}_i\}_{i=1}^N$ normalized such that $k(\boldsymbol{x}_i, \boldsymbol{x}_i) = 1$, let $\boldsymbol{K}_{s,s}$ be the kernel matrix of the SSL dataset. If $\|\Phi(\boldsymbol{x}_a) - \Phi(\boldsymbol{x}_b)\|_{\mathcal{H}} \leq \epsilon$, then $\left\|\boldsymbol{K}_{s,s}^{-1}\boldsymbol{k}_{s,\boldsymbol{x}_a} - \boldsymbol{K}_{s,s}^{-1}\boldsymbol{k}_{s,\boldsymbol{x}_a}\right\| \leq \|\boldsymbol{K}_{s,s}^{-1}\|\sqrt{N}\epsilon$.*

*Proof.* We have that

$$
\begin{aligned}
\left\|\boldsymbol{K}_{s,s}^{-1}\boldsymbol{k}_{s,\boldsymbol{x}_a} - \boldsymbol{K}_{s,s}^{-1}\boldsymbol{k}_{s,\boldsymbol{x}_a}\right\| &\leq \left\|\boldsymbol{K}_{s,s}^{-1}\right\| \left\|\boldsymbol{k}_{s,\boldsymbol{x}_a} - \boldsymbol{k}_{s,\boldsymbol{x}_a}\right\| \\
&= \left\|\boldsymbol{K}_{s,s}^{-1}\right\| \left[\sum_{i=1}^N (\langle\Phi(\boldsymbol{x}_i), \Phi(\boldsymbol{x}_a)\rangle_{\mathcal{H}} - \langle\Phi(\boldsymbol{x}_i), \Phi(\boldsymbol{x}_b)\rangle_{\mathcal{H}})^2\right]^{1/2} \\
&= \left\|\boldsymbol{K}_{s,s}^{-1}\right\| \left[\sum_{i=1}^N (\langle\Phi(\boldsymbol{x}_i), \Phi(\boldsymbol{x}_a) - \Phi(\boldsymbol{x}_b)\rangle_{\mathcal{H}})^2\right]^{1/2} \\
&\leq \left\|\boldsymbol{K}_{s,s}^{-1}\right\| \left[\sum_{i=1}^N \|\Phi(\boldsymbol{x}_i)\|_{\mathcal{H}}^2 \|\Phi(\boldsymbol{x}_a) - \Phi(\boldsymbol{x}_b)\|_{\mathcal{H}}^2\right]^{1/2} \\
&= \left\|\boldsymbol{K}_{s,s}^{-1}\right\| \sqrt{N}\epsilon
\end{aligned}
\tag{43}
$$

$\square$

We are now ready to prove Proposition 3.2.

*Proof.* Note that $\boldsymbol{K}_{s,s}^{-1}\boldsymbol{x}_i = \boldsymbol{e}_i$ where $\boldsymbol{e}_i$ is the vector with a 1 placed on entry $i$ and zeros elsewhere. From equation Equation (7), we have that

$$
\begin{aligned}
k^*(\boldsymbol{x}, \boldsymbol{x}') &= \boldsymbol{k}_{\boldsymbol{x},s}\boldsymbol{K}_{s,s}^{-1} (\boldsymbol{I} + \boldsymbol{A})_+ \boldsymbol{K}_{s,s}^{-1}\boldsymbol{k}_{s,\boldsymbol{x}'} \\
&= (\boldsymbol{K}_{s,s}^{-1}\boldsymbol{k}_{s,x} - \boldsymbol{e}_i + \boldsymbol{e}_i)^{\mathsf{T}} (\boldsymbol{I} + \boldsymbol{A})_+ (\boldsymbol{K}_{s,s}^{-1}\boldsymbol{k}_{s,x'} - \boldsymbol{e}_j + \boldsymbol{e}_j) \\
&= \boldsymbol{e}_i^{\mathsf{T}} (\boldsymbol{I} + \boldsymbol{A})_+ \boldsymbol{e}_j + (\boldsymbol{K}_{s,s}^{-1}\boldsymbol{k}_{s,x} - \boldsymbol{e}_i)^{\mathsf{T}} (\boldsymbol{I} + \boldsymbol{A})_+ \boldsymbol{e}_j + \boldsymbol{e}_i^{\mathsf{T}} (\boldsymbol{I} + \boldsymbol{A})_+ (\boldsymbol{K}_{s,s}^{-1}\boldsymbol{k}_{s,x'} - \boldsymbol{e}_j) \\
&\quad + (\boldsymbol{K}_{s,s}^{-1}\boldsymbol{k}_{s,x} - \boldsymbol{e}_i)^{\mathsf{T}} (\boldsymbol{I} + \boldsymbol{A})_+ (\boldsymbol{K}_{s,s}^{-1}\boldsymbol{k}_{s,x'} - \boldsymbol{e}_j).
\end{aligned}
\tag{44}
$$

Note that $\boldsymbol{e}_i^{\mathsf{T}} (\boldsymbol{I} + \boldsymbol{A})_+ \boldsymbol{e}_j = 1$ since $\boldsymbol{A}_{ij} = 1$. Therefore,

$$
\begin{aligned}
k^*(\boldsymbol{x}, \boldsymbol{x}') &= 1 + (\boldsymbol{K}_{s,s}^{-1}\boldsymbol{k}_{s,x} - \boldsymbol{e}_i)^{\mathsf{T}} (\boldsymbol{I} + \boldsymbol{A})_+ \boldsymbol{e}_j + \boldsymbol{e}_i^{\mathsf{T}} (\boldsymbol{I} + \boldsymbol{A})_+ (\boldsymbol{K}_{s,s}^{-1}\boldsymbol{k}_{s,x'} - \boldsymbol{e}_j) \\
&\quad + (\boldsymbol{K}_{s,s}^{-1}\boldsymbol{k}_{s,x} - \boldsymbol{e}_i)^{\mathsf{T}} (\boldsymbol{I} + \boldsymbol{A})_+ (\boldsymbol{K}_{s,s}^{-1}\boldsymbol{k}_{s,x'} - \boldsymbol{e}_j) \\
&\geq 1 - \|\boldsymbol{K}_{s,s}^{-1}\boldsymbol{k}_{s,x} - \boldsymbol{e}_i\| \|(\boldsymbol{I} + \boldsymbol{A})_+ \boldsymbol{e}_j\| + \|\boldsymbol{K}_{s,s}^{-1}\boldsymbol{k}_{s,x'} - \boldsymbol{e}_j\| \|(\boldsymbol{I} + \boldsymbol{A})_+ \boldsymbol{e}_i\| \\
&\quad + \|\boldsymbol{K}_{s,s}^{-1}\boldsymbol{k}_{s,x} - \boldsymbol{e}_i\| \|(\boldsymbol{I} + \boldsymbol{A})_+\| \|\boldsymbol{K}_{s,s}^{-1}\boldsymbol{k}_{s,x'} - \boldsymbol{e}_j\|.
\end{aligned}
\tag{45}
$$

Let $\|\Phi(\boldsymbol{x}) - \Phi(\boldsymbol{x}_i)\| \leq \epsilon = \frac{\Delta}{5\|\boldsymbol{K}_{s,s}^{-1}\|\sqrt{N}}$ and $\|\Phi(\boldsymbol{x}') - \Phi(\boldsymbol{x}_j)\| \leq \epsilon = \frac{\Delta}{5\|\boldsymbol{K}_{s,s}^{-1}\|\sqrt{N}}$ and by applying Lemma B.3, we have

$$
\begin{aligned}
k^*(\boldsymbol{x}, \boldsymbol{x}') &\geq 1 - \|\boldsymbol{K}_{s,s}^{-1}\boldsymbol{k}_{s,x} - \boldsymbol{e}_i\| \|(\boldsymbol{I} + \boldsymbol{A})_+ \boldsymbol{e}_j\| + \|\boldsymbol{K}_{s,s}^{-1}\boldsymbol{k}_{s,x'} - \boldsymbol{e}_j\| \|(\boldsymbol{I} + \boldsymbol{A})_+ \boldsymbol{e}_i\| \\
&\quad + \|\boldsymbol{K}_{s,s}^{-1}\boldsymbol{k}_{s,x} - \boldsymbol{e}_i\| \|(\boldsymbol{I} + \boldsymbol{A})_+\| \|\boldsymbol{K}_{s,s}^{-1}\boldsymbol{k}_{s,x'} - \boldsymbol{e}_j\| \\
&\geq 1 - 2\sqrt{2}\|\boldsymbol{K}_{s,s}^{-1}\|\sqrt{N}\epsilon - 2\|\boldsymbol{K}_{s,s}^{-1}\|^2 N\epsilon^2 \\
&\geq 1 - (2\sqrt{2} + 2)\|\boldsymbol{K}_{s,s}^{-1}\|\sqrt{N}\epsilon \\
&\geq 1 - 5\|\boldsymbol{K}_{s,s}^{-1}\|\sqrt{N}\epsilon \\
&= 1 - \Delta.
\end{aligned}
\tag{46}
$$

In the above, we used the fact that $\boldsymbol{A}$ is block diagonal with pairwise constraints (see Equation (42)) so $\left\|(\boldsymbol{I} + \boldsymbol{A})_+ \boldsymbol{e}_i\right\| = \sqrt{2}$ and $\left\|(\boldsymbol{I} + \boldsymbol{A})_+\right\| = 2$.

$\square$

## C  IDEALIZATION OF DOWNSTREAM TASKS

In this section, we prove Proposition 3.3 copied below.

**Proposition 3.3** (Ideal SSL outcome). *Given a supervised dataset of $N$ points for binary classification drawn from a distribution with $m_{-1}$ and $m_{+1}$ connected manifolds for classes with labels $-1$ and $+1$ respectively, if the induced kernel matrix of the dataset $\boldsymbol{K}^*$ successfully separates the manifolds such that $k^*(\boldsymbol{x}, \boldsymbol{x}') = 1$ if $\boldsymbol{x}, \boldsymbol{x}'$ are in the same manifold and $k^*(\boldsymbol{x}, \boldsymbol{x}') = 0$ otherwise, then $s_N(\boldsymbol{K}^*) = m_{-1} + m_{+1} = O(1)$.*

*Proof.* There are $m_{+1} + m_{-1}$ total manifolds in the dataset. Assume some ordering of these manifolds from $\{1, 2, \dots, m_{+1} + m_{-1}\}$ and let $\#(i)$ be the number of points in the $i$-th manifold.

The rows and columns of the kernel matrix $\boldsymbol{K}^*$ can be permuted such that it becomes block diagonal with $m_{+1} + m_{-1}$ blocks with block $i$ equal to $\frac{1}{\#(i)}\mathbf{1}_{\#(i)}\mathbf{1}_{\#(i)}^\intercal$ where $\mathbf{1}_k$ is the all ones vector of length $k$. I.e., $\boldsymbol{K}^*$ permuted accordingly takes the form below:

$$\begin{bmatrix} \frac{1}{\#(1)}\mathbf{1}_{\#(1)}\mathbf{1}_{\#(1)}^\intercal & & & \\ & \frac{1}{\#(2)}\mathbf{1}_{\#(2)}\mathbf{1}_{\#(2)}^\intercal & & \\ & & \ddots & \\ & & & \frac{1}{\#(m_{+1}+m_{-1})}\mathbf{1}_{\#(m_{+1}+m_{-1})}\mathbf{1}_{\#(m_{+1}+m_{-1})}^\intercal \end{bmatrix}. \quad (47)$$

Each block of the above is clearly a rank one matrix with eigenvalue 1. Let $\boldsymbol{y}_{m_i} \in \mathbb{R}^{\#(i)}$ be the vector containing all labels for entries in manifold $i$. Then we have

$$\begin{aligned} \boldsymbol{y}^\intercal (\boldsymbol{K}^*)^{-1} \boldsymbol{y} &= \sum_{i=1}^{m_{+1}+m_{-1}} \#(i)^{-1} \left( \langle \mathbf{1}_{\#(i)}, \boldsymbol{y}_{m_i} \rangle \right)^2 \\ &= \sum_{i=1}^{m_{+1}+m_{-1}} \#(i)^{-1} (\sqrt{\#(i)})^2 \\ &= m_{+1} + m_{-1}. \end{aligned} \quad (48)$$

$\square$

### C.1  PROOF OF GENERALIZATION BOUND

For sake of completeness, we include an example proof of the generalization bound referred to in the main text. Common to classic generalization bounds for kernel methods from several prior works (Huang et al., 2021; Meir & Zhang, 2003; Mohri et al., 2018; Vapnik, 1999; Bartlett & Mendelson, 2002), the norm of the linear solution in kernel space correlates with the resulting bound on the generalization error. We closely follow the methodology of (Huang et al., 2021), though other works follow a similar line of reasoning.

Given a linear solution in the reproducing kernel Hilbert space denoted by $\boldsymbol{w}$, we aim to bound the generalization error in the loss function $\ell(y, y') = |\min(1, \max(-1, y)) - y'|$ where $y'$ denotes binary classification targets in $\{-1, 1\}$, $y$ is the output of the kernel function equal to $y = \langle \boldsymbol{w}, \boldsymbol{x} \rangle$ for a corresponding input $\boldsymbol{x}$ in the Hilbert space or feature space. For our purposes, the solution $\boldsymbol{w}$ is given by e.g., Equation (12) equal to

$$\boldsymbol{w} = \sum_{i=1}^{N_t} \left[ \boldsymbol{K}_{t,t}^{*-1} \boldsymbol{y} \right]_i \phi\left( \boldsymbol{x}_i^{(t)} \right), \quad (49)$$

where $\phi(\boldsymbol{x}_i^{(t)})$ denotes the mapping of $\boldsymbol{x}_i^{(t)}$ to the Hilbert space of the kernel. Given this solution, we have that

$$\|\boldsymbol{w}\| = \sqrt{\boldsymbol{y}^\intercal \boldsymbol{K}_{t,t}^{*-1} \boldsymbol{y}}. \quad (50)$$

The norm above controls, in a sense, the complexity of the resulting solution as it appears in the resulting generalization bound.

Given input and output spaces $\mathcal{X}$ and $\mathcal{Y}$ respectively, let $\mathcal{D}$ be a distribution of input/output pairs over the support $\mathcal{X} \times \mathcal{Y}$. Slightly modifying existing generalization bounds via Rademacher complexity arguments (Huang et al., 2021; Bartlett & Mendelson, 2002; Mohri et al., 2018), we prove the following generalization bound.

**Theorem C.1** (Adapted from Section 4.C of (Huang et al., 2021)). *Let $\boldsymbol{x}_1, \ldots, \boldsymbol{x}_N$ and $y_1, \ldots, y_{N_t}$ (with $\boldsymbol{y}$ the corresponding vector storing the scalars $y_i$ as entries) be our training set of $N$ independent samples drawn i.i.d. from $\mathcal{D}$. Let $\boldsymbol{w} = \sqrt{\frac{\mathrm{Tr}(\boldsymbol{K})}{N}} \sum_{i=1}^{N_t} \left[ \boldsymbol{K}^{-1} \boldsymbol{y} \right]_i \phi\left( \boldsymbol{x}_i^{(t)} \right)$ denote the solution to the kernel regression problem normalized by the trace of the data kernel. For an $L$-lipschitz loss function $\ell : \mathcal{Y} \times \mathcal{Y} \to [0, b]$, with probability $1 - \delta$ for any $\delta > 0$, we have*

$$\mathbb{E}_{\boldsymbol{x}, y \sim \mathcal{D}} \left[ \ell(\langle \boldsymbol{w}, \boldsymbol{x} \rangle, y) \right] - \frac{1}{N_t} \sum_{i=1}^{N_t} \ell(\langle \boldsymbol{w}, \boldsymbol{x}_i \rangle, y_i) \leq \frac{2\sqrt{2}L + 3b}{\sqrt{2}} \frac{\sqrt{\mathrm{Tr}(\boldsymbol{K}) \boldsymbol{y}^\intercal \boldsymbol{K}^{-1} \boldsymbol{y}}}{N} + 3b \sqrt{\frac{\log(2/(\delta(e-1)))}{2N}} \tag{51}$$

To prove the above, we apply a helper theorem and lemma copied below.

**Theorem C.2** (Concentration of sum; see Theorem 3.1 in Mohri et al. (2018)). *Let $G$ be a family of functions mapping from $Z$ to $[0, 1]$. Given $N$ independent samples $z_1, \ldots, z_n$ from $Z$, then for any $\delta > 0$, with probability at least $1 - \delta$, the following holds:*

$$\mathbb{E}_z[g(z)] - \frac{1}{N} \sum_{i=1}^{N} g(z_i) \leq 2 \widehat{\mathfrak{R}}_S(G) + 3 \sqrt{\frac{\log(2/\delta)}{2N}}, \tag{52}$$

*where $\widehat{\mathfrak{R}}_S(G)$ denotes the empirical Rademacher complexity of $G$ equal to*

$$\widehat{\mathfrak{R}}_S(G) = \mathbb{E}_\sigma \left[ \sup_{g \in G} \frac{1}{N} \sum_{i=1}^{N} \sigma_i g(z_i) \right], \tag{53}$$

*where $\sigma_i$ are independent uniform random variables over $\{-1, +1\}$.*

**Lemma C.3** (Talagrand's lemma; see Lemma 4.2 in Mohri et al. (2018)). *Let $\Phi : \mathbb{R} \to \mathbb{R}$ be $L$-Lipschitz. Then for any hypothesis set $G$ of real-valued functions,*

$$\widehat{\mathfrak{R}}_S(\Phi \circ G) \leq L \widehat{\mathfrak{R}}_S(G). \tag{54}$$

Now, we are ready to prove Theorem C.1.

*Proof.* We consider a function class $G_\gamma$ defined as the set of linear functions on the reproducing kernel Hilbert space such that $G_\gamma = \{\langle \boldsymbol{w}, \cdot \rangle : \|\boldsymbol{w}\|^2 \leq \gamma\}$. Given a dataset of inputs $\boldsymbol{x}_1, \ldots, \boldsymbol{x}_N$ in the reproducing kernel Hilbert space with corresponding targets $y_1, \ldots, y_N$, let $\epsilon_{\boldsymbol{w}}(\boldsymbol{x}_i) = \ell(\langle \boldsymbol{w}, \boldsymbol{x} \rangle, y_i) \in [0, b]$. The inequality in Theorem C.2 applies for any given $G_\gamma$ but we would like this to hold for all $\gamma \in \{1, 2, 3, \ldots\}$ since $\|\boldsymbol{w}\|$ can be unbounded. Since our loss function $\ell$ is bounded between $[0, b]$, we multiply it by $1/b$ so that it is ranged in $[0, 1]$ as needed for Theorem C.2. Then, we apply Theorem C.2 to $\epsilon_{\boldsymbol{w}}(\boldsymbol{x}_i)$ over the class $G_\gamma$ for each $\gamma \in \{1, 2, 3, \ldots\}$ with probability $\delta_\gamma = \delta(e-1)e^{-\gamma}$. This implies that

$$\mathbb{E}_{\boldsymbol{x}}[\epsilon_{\boldsymbol{w}}(\boldsymbol{x})] - \frac{1}{N} \sum_{i=1}^{N} \epsilon_{\boldsymbol{w}}(\boldsymbol{x}_i) \leq 2 \mathbb{E}_\sigma \left[ \sup_{\|\boldsymbol{v}\|^2 \leq \gamma} \frac{1}{N} \sum_{i=1}^{N} \sigma_i \epsilon_{\boldsymbol{v}}(\boldsymbol{x}_i) \right] + 3b \sqrt{\frac{\log(2/(\delta(e-1))) + \gamma}{2N}}. \tag{55}$$

This shows that for any $\gamma$, the above inequality holds with probability $1 - \delta(e-1)e^{-\gamma}$. However, we need to show this holds for all $\gamma$ simultaneously. To achieve this, we apply a union bound which holds with probability $1 - \sum_{\gamma=1}^{\infty} \delta_\gamma = 1 - \sum_{\gamma=1}^{\infty} \delta(e-1)e^{-\gamma} = 1 - \delta$.

To proceed, we consider the inequality where $\gamma = \lceil \|\boldsymbol{w}\|^2 \rceil$ copied below.

$$\mathbb{E}_{\boldsymbol{x}}[\epsilon_{\boldsymbol{w}}(\boldsymbol{x})] - \frac{1}{N} \sum_{i=1}^{N} \epsilon_{\boldsymbol{w}}(\boldsymbol{x}_i) \leq 2 \mathbb{E}_\sigma \left[ \sup_{\|\boldsymbol{v}\|^2 \leq \lceil \|\boldsymbol{w}\|^2 \rceil} \frac{1}{N} \sum_{i=1}^{N} \sigma_i \epsilon_{\boldsymbol{v}}(\boldsymbol{x}_i) \right] + 3b \sqrt{\frac{\log(2/(\delta(e-1))) + \lceil \|\boldsymbol{w}\|^2 \rceil}{2N}}. \tag{56}$$

Applying Talagrand's lemma (Lemma C.3) followed by the Cauchy-Schwarz inequality,

$$
\begin{aligned}
\mathbb{E}_{\boldsymbol{x}}[\epsilon_{\boldsymbol{w}}(\boldsymbol{x})] - \frac{1}{N}\sum_{i=1}^{N}\epsilon_{\boldsymbol{w}}(\boldsymbol{x}_i) &\leq 2\mathbb{E}_{\sigma}\left[\sup_{\|\boldsymbol{v}\|^2\leq\lceil\|\boldsymbol{w}\|^2\rceil}\frac{1}{N}\sum_{i=1}^{N}\sigma_i\epsilon_{\boldsymbol{v}}(\boldsymbol{x}_i)\right] + 3b\sqrt{\frac{\log(2/(\delta(e-1)))+\lceil\|\boldsymbol{w}\|^2\rceil}{2N}} \\
&\leq 2L\mathbb{E}_{\sigma}\left[\sup_{\|\boldsymbol{v}\|^2\leq\lceil\|\boldsymbol{w}\|^2\rceil}\frac{1}{N}\sum_{i=1}^{N}\sigma_i\langle\boldsymbol{v},\boldsymbol{x}_i\rangle\right] + 3b\sqrt{\frac{\log(2/(\delta(e-1)))+\lceil\|\boldsymbol{w}\|^2\rceil}{2N}} \\
&\leq 2L\mathbb{E}_{\sigma}\left[\sup_{\|\boldsymbol{v}\|^2\leq\lceil\|\boldsymbol{w}\|^2\rceil}\|\boldsymbol{v}\|\left\|\frac{1}{N}\sum_{i=1}^{N}\sigma_i\boldsymbol{x}_i\right\|\right] + 3b\sqrt{\frac{\log(2/(\delta(e-1)))+\lceil\|\boldsymbol{w}\|^2\rceil}{2N}} \\
&\leq 2L\mathbb{E}_{\sigma}\left[\lceil\|\boldsymbol{w}\|\rceil\left\|\frac{1}{N}\sum_{i=1}^{N}\sigma_i\boldsymbol{x}_i\right\|\right] + 3b\sqrt{\frac{\log(2/(\delta(e-1)))+\lceil\|\boldsymbol{w}\|^2\rceil}{2N}}.
\end{aligned}
\tag{57}
$$

Expanding out the quantity $\left\|\frac{1}{N}\sum_{i=1}^{N}\sigma_i\boldsymbol{x}_i\right\|$ and noting that the random variables are independent, we have

$$
\begin{aligned}
\mathbb{E}_{\boldsymbol{x}}[\epsilon_{\boldsymbol{w}}(\boldsymbol{x})] - \frac{1}{N}\sum_{i=1}^{N}\epsilon_{\boldsymbol{w}}(\boldsymbol{x}_i) &\leq 2L\mathbb{E}_{\sigma}\left[\lceil\|\boldsymbol{w}\|\rceil\left\|\frac{1}{N}\sum_{i=1}^{N}\sigma_i\boldsymbol{x}_i\right\|\right] + 3b\sqrt{\frac{\log(2/(\delta(e-1)))+\lceil\|\boldsymbol{w}\|^2\rceil}{2N}} \\
&= \frac{2L}{N}\lceil\|\boldsymbol{w}\|\rceil\mathbb{E}_{\sigma}\left[\sqrt{\sum_{i=1}^{N}\sum_{j=1}^{N}\sigma_i\sigma_j k(\boldsymbol{x}_i,\boldsymbol{x}_j)}\right] + 3b\sqrt{\frac{\log(2/(\delta(e-1)))+\lceil\|\boldsymbol{w}\|^2\rceil}{2N}} \\
&\leq \frac{2L}{N}\lceil\|\boldsymbol{w}\|\rceil\sqrt{\mathbb{E}_{\sigma}\left[\sum_{i=1}^{N}\sum_{j=1}^{N}\sigma_i\sigma_j k(\boldsymbol{x}_i,\boldsymbol{x}_j)\right]} + 3b\sqrt{\frac{\log(2/(\delta(e-1)))+\lceil\|\boldsymbol{w}\|^2\rceil}{2N}} \\
&= \frac{2L}{N}\sqrt{\sum_{i=1}^{N}k(\boldsymbol{x}_i,\boldsymbol{x}_i)} + 3b\sqrt{\frac{\log(2/(\delta(e-1)))+\lceil\|\boldsymbol{w}\|^2\rceil}{2N}} \\
&= \frac{2L}{N}\lceil\|\boldsymbol{w}\|\rceil\sqrt{\mathrm{Tr}(\boldsymbol{K})} + 3b\sqrt{\frac{\log(2/(\delta(e-1)))+\lceil\|\boldsymbol{w}\|^2\rceil}{2N}}.
\end{aligned}
\tag{58}
$$

Since we normalize such that $\mathrm{Tr}(\boldsymbol{K}) = N$, we have

$$
\begin{aligned}
\mathbb{E}_{\boldsymbol{x}}[\epsilon_{\boldsymbol{w}}(\boldsymbol{x})] - \frac{1}{N}\sum_{i=1}^{N}\epsilon_{\boldsymbol{w}}(\boldsymbol{x}_i) &\leq \frac{2L}{\sqrt{N}}\lceil\|\boldsymbol{w}\|\rceil + 3b\sqrt{\frac{\log(2/(\delta(e-1)))+\lceil\|\boldsymbol{w}\|^2\rceil}{2N}} \\
&\leq \frac{2L}{\sqrt{N}}\lceil\|\boldsymbol{w}\|\rceil + 3b\sqrt{\frac{\log(2/(\delta(e-1)))}{2N}} + 3b\frac{\lceil\|\boldsymbol{w}\|\rceil}{\sqrt{2N}} \\
&= \frac{2\sqrt{2}L+3b}{\sqrt{2N}}\lceil\|\boldsymbol{w}\|\rceil + 3b\sqrt{\frac{\log(2/(\delta(e-1)))}{2N}}.
\end{aligned}
\tag{59}
$$

Plugging in $\|\boldsymbol{w}\|^2 = \boldsymbol{y}^{\intercal}\boldsymbol{K}^{-1}\boldsymbol{y}$ and noting that we normalized the kernel to have $\mathrm{Tr}(\boldsymbol{K}) = N$ thus completes the proof.

$\square$

# D NUMERICAL EXPERIMENTS

All of our implementations are based on the common Python scientific library Numpy/Scipy (Harris et al., 2020) and runs on CPU (no GPU used). Kernel algorithms were performed using the Scikit-learn package (Pedregosa et al., 2011). Hyperparameters were chosen via a grid search over the kernel algorithm parameters (e.g., regularization terms) and the loss function hyperparameters where appropriate. For calculation of the neural tangent kernel, we used the neuraltangents package (Novak et al., 2020).

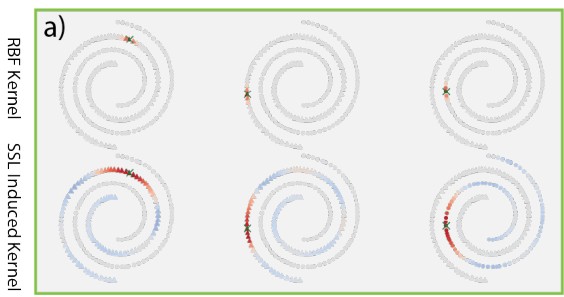 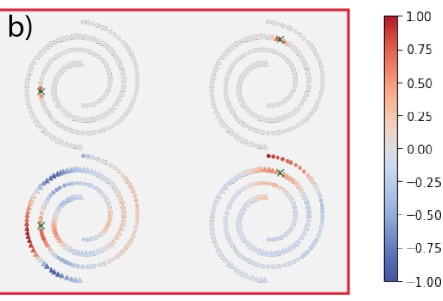

Figure 5: Inner products comparison in the RBF kernel space (first row) and induced kernel space (second row). The induced kernel is computed based on Equation 7 and the graph adjacency matrix $A$ is derived from the inner product neighborhood in the RBF kernel space, i.e., using the neighborhoods as data augmentation. We plot three randomly chosen points' kernel entries with respect to the other points on the manifolds. a) When the neighborhood augmentation range used to construct the adjacency matrix is small enough, the SSL-induced kernel faithfully learns the topology of the entangled spiral manifolds. b) When the neighborhood augmentation range used to construct the adjacency matrix is too large, it creates the "short-circuit" effect in the induced kernel space. Each subplot on the second row is normalized by dividing its largest absolute value for better contrast.

Here, we provide the contrastive SSL-induced kernel visualization in Figure 5 to show how the SSL-induced kernel is helping to manipulate the distance and disentangle manifolds in the representation space. In Figure 5, we study two entangled 1-D spiral manifolds in a 2D space with 200 training points uniformly distributed on the spiral manifolds. We use the contrastive SSL-induced kernel, following Equation 7, to demonstrate this result, whereas the non-contrastive SSL-induced kernel is provided earlier in the main text. We use the radial basis function (RBF) kernel to calculate the inner products between different points and plot a few points' RBF neighborhoods in the first row of Figure 2, i.e., $k(x_1, x_2) = \exp(-\frac{\|x_1-x_2\|^2}{2\sigma^2})$. As we can see, the RBF kernel captures the locality of the 2D space. Next, we use the RBF kernel space neighborhoods to construct the adjacency matrix $A$ and any training points with $k(x_1, x_2) > d$ are treated as connected vertices, i.e., $A_{ij} = 1$ and $A_{ij} = 0$ otherwise. The diagonal entries of $A$ are 0. This construction can be considered as using the Euclidean neighborhoods of $x$ as the data augmentation. In the second row of Figure 5, we show the selected points' inner products with the other training points in the SSL-induced kernel space. Given $\sigma$, when $d$ is chosen small enough, we can see in the second row of Figure 5(a) that the SSL-induced kernel faithfully captures the topology of manifolds and leads to a more disentangled representation. Figure 5(b) shows that when $d$ is too large, i.e., an improper data augmentation, the SSL-induced kernel leads to the "short-circuit" effect, and the two manifolds are mixed in the representation space.

## D.2 TIME SERIES DATA AND CIFAR10

We also add in table 1 additional empirical validation on a few time-series datasets extracted from UCR (Chen et al., 2015) as well as a few controlled experiments on CIFAR10. For the latter we propose the same set-up as in our MNIST experiments: i.e., translation and rotation data-augmentation. For the UCR experiments, we implement translations and add white noise as data-augmentation. Surprisingly, despite such simple augmentations, the SSL induced kernel achieves similar performance to the supervised baselines.

## D.3 MNIST WITH NEURAL TANGENT KERNELS

In further exploring the performance of SSL kernel methods on small datasets, we perform further numerical experiments on the MNIST dataset using kernel functions derived from the neural tangent kernel (NTK) of commonly used neural networks (Jacot et al., 2018). We use the neural-tangents package to explicitly calculate the NTK for a given architecture (Novak et al., 2020). The basic setup is repeated from Section 4.2 where two different types of augmentations are performed on im-

| Method training size/#DA | CIFAR10 100/20 | CIFAR10 100/30 | CIFAR10 200/20 | CIFAR10 300/20 | Japanese Vowels | ECG200 | ECGFiveDays |
|---|---|---|---|---|---|---|---|
| Original kernel (w/ DA) | 23.07 | 22.99 | 29.33 | 30.65 | 85.13 | 70.00 | 53.07 |
| Original kernel (no DA) | 21.49 | 21.49 | 26.81 | 29.16 | 84.86 | 64.00 | 52.49 |
| SSL | 22.68 | 23.62 | 27.52 | 29.53 | 93.78 | 68.00 | 52.84 |

Table 1: Test set accuracy on various datasets comparing the different methods. For CIFAR10, the same augmentation was used as for MNIST (rotation + translation) while for the time-series dataset, translation and white noise was used. Every implementation uses the same methodology and hyperparameter tuning as that of the fig. 3 experiment.

| | | | Test Set Accuracy | | | | | | |
| | | Num. Augmentations | 16 | 32 | 64 | 128 | 256 | 512 | 1024 |
| Augmentation | Samples | Algorithm | | | | | | | |
|---|---|---|---|---|---|---|---|---|---|
| Gaussian blur | 16 | self-supervised | 32.3 | 32.3 | 32.6 | 29.1 | 32.8 | 32.3 | 32.5 |
| | | baseline (no augment) | 21.8 | 21.8 | 21.8 | 21.8 | 21.8 | 21.8 | 21.8 |
| | | baseline (augmented) | 34.8 | 34.9 | 35.0 | 35.0 | 35.0 | 35.0 | 35.0 |
| | 64 | self-supervised | 71.8 | 71.8 | 71.6 | 71.4 | 72.1 | | |
| | | baseline (no augment) | 65.0 | 65.0 | 65.0 | 65.0 | 65.0 | | |
| | | baseline (augmented) | 74.6 | 74.6 | 74.7 | 74.7 | 74.7 | | |
| | 256 | self-supervised | 87.5 | 88.0 | 88.0 | | | | |
| | | baseline (no augment) | 86.5 | 86.5 | 86.5 | | | | |
| | | baseline (augmented) | 88.6 | 88.6 | 88.6 | | | | |
| translate, zoom rotate | 16 | self-supervised | 34.2 | 35.2 | 36.0 | 36.8 | 37.3 | 37.5 | 38.0 |
| | | baseline (no augment) | 21.8 | 21.8 | 21.8 | 21.8 | 21.8 | 21.8 | 21.8 |
| | | baseline (augmented) | 37.5 | 38.5 | 39.3 | 40.2 | 41.1 | 41.5 | 41.7 |
| | 64 | self-supervised | 73.4 | 74.8 | 75.3 | 75.9 | 76.8 | | |
| | | baseline (no augment) | 65.0 | 65.0 | 65.0 | 65.0 | 65.0 | | |
| | | baseline (augmented) | 77.7 | 78.8 | 79.4 | 79.7 | 80.0 | | |
| | 256 | self-supervised | 90.4 | 91.0 | 91.5 | | | | |
| | | baseline (no augment) | 86.5 | 86.5 | 86.5 | | | | |
| | | baseline (augmented) | 90.4 | 90.8 | 91.1 | | | | |

Table 2: **NTK for 3-layer fully connected network:** Test set accuracy of SVM using the neural tangent kernel of a 3 layer fully connected network in only a supervised (baseline) setting or via the induced kernel in a self-supervised setting. The induced kernel for the SSL algorithm is calculated using the contrastive induced kernel. Numbers shown above are the test set accuracy for classifying MNIST digits for small dataset sizes with the given number of samples. Due to the quadratic scaling of memory and runtime for kernel methods, we restricted analysis to more feasible settings where there were less than 25,000 total samples (number of augmentations times number of samples).

ages. As before, we consider augmentations by Gaussian blurring of the pixels or small translations, rotations, and zooms.

Test set accuracy results are shown in Table 2 for the NTK associated to a 3-layer fully connected network with infinite width at each hidden layer. Similarly, in Table 3, we show a similar analysis for the NTK of a CNN with seven convolutional layers followed by a global pooling layer. We use the contrastive induced kernel for the SSL kernel algorithm. The findings are similar to those reported in the main text. The SSL method performs similarly to the baseline method without any self-supervised learning but including labeled augmented datapoints in the training set. In some cases, the SSL method even outperforms the baseline augmented setting.

### D.4 ANALYSIS OF DOWNSTREAM SOLUTION

To empirically analyze generalization, we track the complexity quantity $s_N(\boldsymbol{K})$ here defined in Equation (13) and copied below:

$$s_N(\boldsymbol{K}) = \frac{\mathrm{Tr}(\boldsymbol{K})}{N} \boldsymbol{y}^\intercal \boldsymbol{K}^{-1} \boldsymbol{y}, \tag{60}$$

| Augmentation | Samples | Num. Augmentations Algorithm | Test Set Accuracy 16 | 32 | 64 | 128 | 256 | 512 | 1024 |
|---|---|---|---|---|---|---|---|---|---|
| Gaussian blur | 16 | self-supervised | 28.4 | 28.4 | 28.4 | 28.4 | 28.3 | 28.3 | 28.4 |
| | | baseline (no augment) | 25.6 | 25.6 | 25.6 | 25.6 | 25.6 | 25.6 | 25.6 |
| | | baseline (augmented) | 26.6 | 26.6 | 26.6 | 26.6 | 26.6 | 26.6 | 26.6 |
| | 64 | self-supervised | 60.0 | 60.0 | 60.1 | 60.1 | 60.1 | | |
| | | baseline (no augment) | 55.6 | 55.6 | 55.6 | 55.6 | 55.6 | | |
| | | baseline (augmented) | 56.5 | 56.5 | 56.5 | 56.5 | 56.5 | | |
| | 256 | self-supervised | 87.6 | 87.9 | 87.6 | | | | |
| | | baseline (no augment) | 81.5 | 81.5 | 81.5 | | | | |
| | | baseline (augmented) | 82.9 | 82.9 | 82.9 | | | | |
| translate, zoom rotate | 16 | self-supervised | 33.4 | 33.8 | 34.4 | 34.6 | 35.1 | 35.4 | 35.3 |
| | | baseline (no augment) | 25.6 | 25.6 | 25.6 | 25.6 | 25.6 | 25.6 | 25.6 |
| | | baseline (augmented) | 30.6 | 30.7 | 30.8 | 31.6 | 31.7 | 31.9 | 32.0 |
| | 64 | self-supervised | 70.7 | 72.7 | 73.2 | 73.8 | 74.3 | | |
| | | baseline (no augment) | 55.6 | 55.6 | 55.6 | 55.6 | 55.6 | | |
| | | baseline (augmented) | 66.0 | 67.8 | 68.8 | 69.5 | 70.2 | | |
| | 256 | self-supervised | 91.9 | 92.3 | 92.6 | | | | |
| | | baseline (no augment) | 81.5 | 81.5 | 81.5 | | | | |
| | | baseline (augmented) | 89.5 | 90.0 | 90.3 | | | | |

Table 3: **NTK for 7-layer convolutional neural network:** Test set accuracy of SVM using the neural tangent kernel of a CNN with 7 layers of $3 \times 3$ convolution followed by a global pooling layer. The induced kernel for the SSL algorithm is calculated using the contrastive induced kernel and compared to the standard SVM using the kernel itself with or without augmentation. Numbers shown above are the test set accuracy for classifying MNIST digits for small dataset sizes with the given number of samples. Due to the quadratic scaling of memory and runtime for kernel methods, we restricted analysis to more feasible settings where there were less than 25,000 total samples (number of augmentations times number of samples).

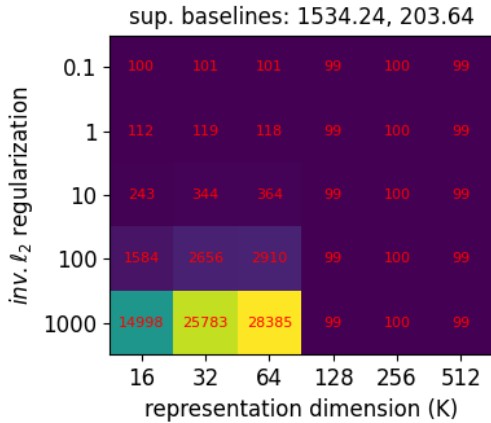

Figure 6: Depiction of $s_N(\boldsymbol{K})$ (Equation (13)) for the case of MNIST classification as depicted in the left figure in Figure 4 computed on the training set. This setting considers MNIST classification with the contrastive kernel on a 10000-sample dataset with 100 original MNIST samples and 100 augmentations. $s_N(\boldsymbol{K})$ for the supervised baselines for the full dataset (including labeled augmented samples) and only the original dataset (no augmented samples) are calculated as approximately 1534 and 203 respectively. Many values of $s_N(\boldsymbol{K})$ are smaller than the baseline numbers especially at points where the test set accuracy for the SSL induced kernel was comparitively larger. Here, we recover the trend of the test set performances obtained from Figure 4 showing that $s_N(\boldsymbol{K})$ is potentially a good indicator of test accuracy.

where $\boldsymbol{y}$ is a vector of targets and $\boldsymbol{K}$ is the kernel matrix of the supervised dataset. As a reminder, the generalization gap can be bounded with high probability by $O(\sqrt{s_N(\boldsymbol{K})/N})$, and in the main text, we provided heuristic indication that this quantity may be reduced when using the induced kernel. Here, we empirically analyze whether this holds true in the MNIST setting considered in the main text. As shown in Figure 6, the SSL induced kernel produces smaller values of $s_N(\boldsymbol{K})$ than its corresponding supervised counterpart. We calculate this figure by splitting the classes into binary parts (even and odd integers) in order to construct a vector $\boldsymbol{y}$ that mimics a binary classification task. Comparing this to the test accuracy results reported in Figure 4, it is clear that $s_N(\boldsymbol{K})$ also correlates

inversely with test accuracy as expected. The results here lend further evidence to the hypothesis that the induced kernel better correlates points along a data manifold as outlined in Proposition 3.3.

