# OpenReview forum: "Joint Embedding Self-Supervised Learning in the Kernel Regime"
_ICLR.cc/2023/Conference — Submitted to ICLR 2023_

### Official Review · Reviewer_FCEZ · 2022-10-22

**Confidence:** 5
**Correctness:** 3
**Technical Novelty And Significance:** 3
**Empirical Novelty And Significance:** 3
**Recommendation:** 6

**Clarity, Quality, Novelty And Reproducibility:**

The description of this paper is clear, the derivation process is detailed, and it is novel, but the complex process and design make me worry about its reproducibility.

**Strength And Weaknesses:**

Strengths:
1、The theoretical derivation is rigorous;
2、The contribution description is relatively clear;
3、The writing process is clear.

Weaknesses:
1. Please describe the motivation of the proposed method in detail in the abstract;
2. For the organization of other parts of this article, it should be directly specific to the chapter, rather than described in sequence;
3. In the experimental part, the equipment type used and some hyper-parameter settings (the specific usage of the data set) should be described in detail;
4. The experimental part only proves the effectiveness of the proposed method, and does not compare with other advanced algorithms;
5. The design concept of the paper is complicated and cumbersome, please prove its reproducibility;
6. The last sentence of the paragraph above equation (14) lacks punctuation;
7. Tables are also a language. Don't just mention that the results of your proposed method are better than others, but try to explain why.


**Summary Of The Paper:**

Modern methods in SSL form representations based on known or constructed relationships between samples and are particularly effective in this task. Here, this paper aim to extend this framework to incorporate algorithms based on kernel methods, where the embeddings are constructed from linear maps acting on the kernel feature space.

**Summary Of The Review:**

1、For a contrastive and non-contrastive loss, this paper provide closed form solutions when the algorithm is trained over a single batch of data.
2、This paper show that a version of the representer theorem in kernel methods can be used to formulate kernelized SSL tasks as optimization problems.

---

> ### Author Response · Authors · 2022-11-09
> **Initial response to reviewer**
>
> We thank the reviewer for their nice comments about the theoretical rigor and writing quality of our paper. We also thank the reviewer for their helpful comments and feedback on our manuscript. As for the other reviewers, we have taken the many comments into account and wish to respond to comments as soon as possible to give reviewers the opportunity to reply if they so choose. We have updated the paper where appropriate and are now working on new experiments to include before the deadline. We have indicated in our response to this reviewer and other reviewers what changes have been made and what experiments will be added.
>
>
> > Please describe the motivation of the proposed method in detail in the abstract;
>
> We have amended the abstract to make the motivation clearer.
>
> > For the organization of other parts of this article, it should be directly specific to the chapter, rather than described in sequence;
>
> We apologize for our misunderstanding, but can you expand on what you mean by “directly specific to the chapter”.
>
> > In the experimental part, the equipment type used and some hyper-parameter settings (the specific usage of the data set) should be described in detail;
>
> We thank the author for pointing out the specifics needed. We have added these details in the upcoming draft. Also, we have shared the code with the reviewers and many of these details are included there (but will nevertheless also be included in the upcoming draft).
>
> > The experimental part only proves the effectiveness of the proposed method, and does not compare with other advanced algorithms;
>
> Can the reviewer let us know what algorithms they have in mind?  We wanted to focus this paper on SSL specifically in relation to the baseline supervised algorithm. As mentioned in the response to other reviewers, there are a host of kernel algorithms for learning on graphs/manifolds that have connections to our works, but since our central goal was not to “beat” these algorithms or produce a new optimal kernel algorithm, we did not focus on this setting. Instead, we aimed to focus on the SSL task vs. the baseline algorithm as is commonly done in deep network settings to gain insights into the training and help translate that commonly employed setting into the kernel regime. Please let us know if you believe that more experiments/analysis is needed and we are happy to consider those.
>
> > The design concept of the paper is complicated and cumbersome, please prove its reproducibility;
>
> We apologize, but we cannot understand this comment. Can the reviewer please expand on what they mean by the design concept of the paper?
>
> > The last sentence of the paragraph above equation (14) lacks punctuation;
>
> Thank you for pointing this out. We have corrected this typo.
>
> > Tables are also a language. Don't just mention that the results of your proposed method are better than others, but try to explain why.
>
> If we understand the reviewer correctly, they are expressing that we should add more details about why or when our method outperforms baseline methods. If our understanding is correct, then we will add more details on this point in our experiments to show the role of SSL in improving performance. Of course, this is a central question that still has no complete answer and we hope that future work can add further insights to this point.

---

### Official Review · Reviewer_qCTa · 2022-10-23

**Confidence:** 3
**Correctness:** 4
**Technical Novelty And Significance:** 3
**Empirical Novelty And Significance:** 3
**Recommendation:** 5

**Clarity, Quality, Novelty And Reproducibility:**

The paper is not hard to follow. Since the applicability of theoretical results seems relatively narrow, I think that the paper does not have an impact that much.

**Strength And Weaknesses:**

Strength
- The paper considers SSL with a linear map on an RKHS. Due to the restriction of the problem setting, the authors provide simple theoretical results that promote an intuitive understanding of SSL. The problem falls into a simple optimization problem. An explicit form of the solution is presented for non-contrastive/contrastive loss.

Weaknesses
- Nowadays, SSL is widely used for learning tasks with large data sets. Considering such a situation, the scope of this paper is relatively narrow. This paper focuses on SSL with kernel features. Though Proposition 3.1 deals with a general class of loss functions, the theoretical analysis is provided only for VICReg loss and spectral contrastive loss.
- In analyzing the generalization performance of downstream tasks, the complexity quantity s_N(K) is considered. It is unclear how the complexity quantity relates to the concept of alignment and uniformity [1], which are regarded as important features in SSL.[1] Wang, and Isola, Understanding Contrastive Representation Learning through Alignment and Uniformity on the Hypersphere, ICML,2020.
- In numerical experiments, feature extraction with RBF kernel is reported. However, most readers will be interested in feature extraction using deep neural networks (DNNs). The authors showed the SSL with neural tangent kernel (NTK) regime in the appendix. The authors should conduct intensive numerical studies using DNN and investigate how much their theoretical findings for the NTK regime explain numerical results. Though the authors discussed it in section 5, the numerical results in the present paper are not very informative in understanding the usefulness of SSL in practice.


**Summary Of The Paper:**

This paper studies self-supervised learning (SSL) with kernels. The authors prove the representer theorem for SSL and provide explicit expressions of the optimal induced kernel for some SSL methods using contrastive or non-contrastive loss functions. The prediction accuracy on downstream tasks is analyzed based on the complexity quantity related to kernels. In numerical experiments, the authors investigate hyper-parameter tuning and the relation between data augmentation and generalization property.

**Summary Of The Review:**

Though the theoretical results in the paper are rigid, the scope of the paper seems narrow.

---

> ### Author Response · Authors · 2022-11-09
> **Initial response to reviewer**
>
> We thank the reviewer for their comments and feedback on our manuscript. We have taken the many comments into account and updated the draft to account for these comments. We also are preparing additional experiments as stated below, but we wish to respond to comments as soon as possible to give reviewers the opportunity to reply if they so choose. We will again update the draft once those experiments are complete.
>
> Responses to individual comments below:
> > Nowadays, SSL is widely used for learning tasks with large data sets. Considering such a situation, the scope of this paper is relatively narrow. This paper focuses on SSL with kernel features. Though Proposition 3.1 deals with a general class of loss functions, the theoretical analysis is provided only for VICReg loss and spectral contrastive loss.
>
> We focused on the VICReg and spectral contrastive loss since they offered the cleanest analysis with closed-form solutions and extended nicely to frame the general theoretical picture. As the reviewer states, proposition 3.1 does extend to other loss functions in practice. It is still an open question which ones admit closed form solutions (we were unable to find more closed form solutions in our work but there are likely more!). Nevertheless, to make the situation clearer, we have explicitly stated this fact and also added details in the appendix on how to extend our framework to cover other loss functions.
>
> > In analyzing the generalization performance of downstream tasks, the complexity quantity s_N(K) is considered. It is unclear how the complexity quantity relates to the concept of alignment and uniformity [1], which are regarded as important features in SSL.[1] Wang, and Isola, Understanding Contrastive Representation Learning through Alignment and Uniformity on the Hypersphere, ICML,2020.
>
> We agree that uniformity and alignment are important properties needed for SSL to work. The complexity quantity is actually strongly tied to alignment which becomes crucial for the downstream task. In fact, the complexity quantity is minimized when the data is most aligned on classes or manifolds of the data as we state in the text. Furthermore, the optimal solutions output orthogonal functions into the representation dimension of size $K$ which are in a sense uniform from their orthogonality properties. Thus, there is definitely a clear linkage between our setting and that of Wang and Isola; however, one important distinction here is that the complexity quantity s_N(K) is defined for downstream classification tasks, whereas the quantities of alignment and uniformity are calculated on the SSL task. Despite this distinction, we agree that we could have made this clearer and we thank the reviewer for raising this point. We have stated this connection in the main text.
>
> > In numerical experiments, feature extraction with RBF kernel is reported. However, most readers will be interested in feature extraction using deep neural networks (DNNs). The authors showed the SSL with neural tangent kernel (NTK) regime in the appendix. The authors should conduct intensive numerical studies using DNN and investigate how much their theoretical findings for the NTK regime explain numerical results. Though the authors discussed it in section 5, the numerical results in the present paper are not very informative in understanding the usefulness of SSL in practice.
>
> In this comment, we believe that the reviewer is asking us to determine how close training of NTK via kernel methods is to training of finite width deep networks with the corresponding SSL loss network. If we have misunderstood the reviewer, then please let us know.
>
> We share the reviewer’s desire but do not feel this is the right place for this analysis. This is an important and useful task, but one that would require many choices and potential approximations to make such a comparison feasible. For example, consider the following three points. First, kernel methods cannot be trained on nearly as many datapoints (orders of magnitude difference) as their neural network counterparts. Second, understanding the “features” of a NTK is already a challenging task which in our setting would require further incorporation into the SSL optimization. Third, we would potentially have to incorporate some “early stopping” criterion for the kernel method to match finite time SGD on a neural network. We share these points not to distract from the reviewer’s concern which is one we agree on. Instead, we want to point out that such an analysis would require significant care and detail to carry through and we felt it would distract from the general framework and idea of the paper. Nevertheless, we do hope to do such work in the future and believe this is a very important area to explore in more detail and have as a separate work.

---

### Official Review · Reviewer_Gmj4 · 2022-10-25

**Confidence:** 4
**Correctness:** 4
**Technical Novelty And Significance:** 3
**Empirical Novelty And Significance:** 2
**Recommendation:** 3

**Clarity, Quality, Novelty And Reproducibility:**

The paper is clear and good quality development of the approach and formulations, but the message of the paper is not very clear and seems like the intro and motivation do not match the body and experiments of the paper.  I think the idea of considering modern SSL loss using kernel representations is novel - so that is original - but the particular idea and formulation seems straight forward and inline with past work on kernel learning on graphs and manifolds.

Reproducibility as high as code is provided.

**Strength And Weaknesses:**

Strengths:
1. It is an interesting idea and direction to consider contrastive / SSL representation learning approaches from the perspective of kernel learning.

2. Several useful properties are proven about the SSL kernel formulation, and solutions derived (including closed-form) for SSL formulations / objectives - which could be useful to researchers and practitioners.

3. Analysis and experiments were formed to better understand ad explain the SSL induced kernels.

Weaknesses:
1.  The paper feels a bit incomplete - the properties developed are not unexpected, and the intro argues for understanding of common modern SSL representation learning methods.  However, the paper really just presents a kernel algorithm for SSL and there does not really seem to be a connection to deep learning based SSL learning methods.

Essentially it's not really clear what is the real goal or objective of the paper.  If it's to propose a new SSL representation learning algorithm using kernel representations, then it needs more experiment evaluation to demonsrate its benefits over existing approaches.  If instead it's to reveal insights about current SSL representation learning approaches and losses used - this was also not accomplished.  As such, it doesn't feel complete.

2. Experiments seem lacking.  Comparison with more methods and more datasets would be better.  As mentioned above, if the point is a new kernel SSL algorithm - more thorough comparison with other kernel algorithms and with existing representation learning algorithms on multiple datasets and tasks should be done - not just a simple classification task on EMNIST and MNIST comparing only to using the standard SVM classifier with RBF kernel.   If the point is to understand modern SSL algorithms, experiments using those algorithms and showing how the proposed method helps understand their results should be included.

Additionally the classification results do not seem very convincing for the benefit of using the proposed approach - in particular if I understand it correctly, Figure 3 show that the accuracy by simply using augmented training data as opposed to the induced kernel based on the augmentations consistently yields higher test accuracy.


3. The results / resulting formulations and properties are not very surprising / seem somewhat expected, including the method itself, and seem similar and closely related to much prior work on learning on graphs or manifolds (using kernels).  It's not completely clear to me how this brings much novelty beyond past kernel approaches to learning across a graph / manifold - as these use the graph laplacian / manifold assumption in a similar way.  I think more discussion and comparison is warranted.  Baselines for comparing prediction results for a relationship-informed kernel learning representation approach should include such laplacian / manifold informed versions of the corresponding estimators, like graph / manifold regularized SVM.

Examples of such work:
- "Kernels and Regularization on Graphs" Smola and Kondor.
- "Learning on Graph with Laplacian Regularization" Ando and Zhang
- "Manifold Regularization: A Geometric Framework for Learning from Labeled and Unlabeled Examples" Belkin et al.


4. Minor comments:
- In the introduction, one type of self supervised learning missing is the other major class of SSL not mentioned - predicting held-out parts of the input data (as in transformer models).
- Figure 1 does not seem very helpful to me and doesn't seem to explain anything.



**Summary Of The Paper:**

The authors consider "joint embedding" self-supervised learning (SSL) formulations applied to a kernel learning setting.  That is, finding / learning a linear mapping in the feature space (induced by some kernel) that minimizes commonly used SSL learning objectives, such as for contrastive learning.

The authors prove properties about this mapping function and the solution to such problems, and derive closed form optimal solutions to specific formulations (objectives), as well as a semi definite programming formulation to solve a general objective over batches.

The authors experimentally explore the SSL kernel induced kernels - comparing the induced kernel from the SSL formulation (which leverages an adjacency matrix as part of fitting the kernel space mapping / the SSL objective) with the base kernel (RBF kernels used with NTK in the appendix).  They compare qualitatively by plotting the similar pairs under each kernel on the spiral toy dataset, and compare classification accuracy and for varying hyper parameters / settings.

**Summary Of The Review:**

While the development of the properties and formulations, and closed form solutions for this approach to learning and induced SSL kernel could be useful, overall I feel more like the paper is incomplete and lacks a coherent purpose and objective to achieve, and so falls short.  I think it could benefit from more experiments and comparisons, and elucidating the connection with current SSL approaches and what it reveals about them (as motivated in the introduction).

---

> ### Author Response · Authors · 2022-11-09
> **Initial response to reviewer**
>
> We thank the reviewer for their in-depth review and their many helpful comments about our paper. We also thank the reviewer for commending the presentation and writing quality of the paper, and hope that we can address their concerns about the unclear message of the paper both in our response below and in upcoming changes to the draft.
>
> To answer the reviewer's comments as soon as possible, we are providing a preliminary response now to give the reviewer the opportunity to respond if they so choose. We have updated the manuscript where appropriate and are concurrently adding experiments before the deadline in response to the reviewer’s insightful remarks. We will provide updates once those experiments are complete. We have indicated below exactly what changes will be made.
>
> Responses to individual comments below:
>
> > 1. The paper feels a bit incomplete - the properties developed are not unexpected, and the intro argues for understanding of common modern SSL representation learning methods. However, the paper really just presents a kernel algorithm for SSL and there does not really seem to be a connection to deep learning based SSL learning methods.
> Essentially it's not really clear what is the real goal or objective of the paper. If it's to propose a new SSL representation learning algorithm using kernel representations, then it needs more experiment evaluation to demonsrate its benefits over existing approaches. If instead it's to reveal insights about current SSL representation learning approaches and losses used - this was also not accomplished. As such, it doesn't feel complete.
>
> We thank the reviewer for pointing out their concerns about the clarity of the objective. Our main objective was to demonstrate that commonly employed SSL algorithms admit a nice formulation in the kernel regime that could be investigated both independently and as a model to examine deep network characteristics in SSL. As the reviewer rightly points out and as indicated in our text, kernel learning over graphs has a rich history of study, so our primary aim was not to supplant or “beat” this setting with a new algorithm. Instead, we aimed to introduce a nice mathematical setting where SSL could be analyzed in the kernel regime. We agree with the reviewer that we could have pointed this out more clearly and have explicitly stated the above in our updated draft.
>
>
> > 2. Experiments seem lacking. Comparison with more methods and more datasets would be better. As mentioned above, if the point is a new kernel SSL algorithm - more thorough comparison with other kernel algorithms and with existing representation learning algorithms on multiple datasets and tasks should be done - not just a simple classification task on EMNIST and MNIST comparing only to using the standard SVM classifier with RBF kernel. If the point is to understand modern SSL algorithms, experiments using those algorithms and showing how the proposed method helps understand their results should be included.
>
> We will add experiments with more datasets (CIFAR and time-series data) and more baseline comparisons (including ridge regression and SVM). Furthermore, we will add some comments about common features of training the kernel vs. standard deep network architectures (e.g., hyperparameter tuning similarities). We chose not to include experiments comparing to other graph or manifold embedding kernel approaches since as stated earlier, our goal was not to supplant or challenge these algorithms. In fact, we do not expect the SSL kernel algorithms to beat the algorithms custom designed for kernel methods so we did not want to distract from the main message of the paper. We of course welcome any comments on this point.
>
>
> > Additionally the classification results do not seem very convincing for the benefit of using the proposed approach - in particular if I understand it correctly, Figure 3 show that the accuracy by simply using augmented training data as opposed to the induced kernel based on the augmentations consistently yields higher test accuracy.
>
> The performance depends on the setting. We found some cases where the induced kernel outperforms the augmented dataset with standard kernel algorithms. This performance also clearly depended on the form of the data augmentation. With correct data augmentation (those aligned well with the dataset), we found situations where SSL would outperform baseline methods. Once we finish further experiments on more datasets (as stated earlier), we will add additional analysis on this point.
>
>
> (1/2)

---

> > ### Author Response · Authors · 2022-11-09
> > **Initial response to reviewer (cont.)**
> >
> > > 3. The results / resulting formulations and properties are not very surprising / seem somewhat expected, including the method itself, and seem similar and closely related to much prior work on learning on graphs or manifolds (using kernels). It's not completely clear to me how this brings much novelty beyond past kernel approaches to learning across a graph / manifold - as these use the graph laplacian / manifold assumption in a similar way. I think more discussion and comparison is warranted.
> >
> > We agree with the reviewer that our results are related to prior work on learning on graphs and manifolds, and we were inspired by much of that work. As previously mentioned, our aim was to translate the concepts into the SSL environment employed in contemporary deep learning tasks rather than to outperform methods derived from these publications. As far as we are aware, no works had studied this specifically for the SSL algorithms and loss functions that are commonly used in deep learning settings. Our derivations and analyses, we believe, were justified by this, but nevertheless, the reviewer’s point is well taken. To help address this concern, we have expanded the related works (also in the appendix) to include more details about the many works that inspired and motivated this one from a kernel perspective. Throughout the text, we have also added more sentences and remarks connecting the ideas in this work to those of previous works where appropriate.
> >
> > > Baselines for comparing prediction results for a relationship-informed kernel learning representation approach should include such laplacian / manifold informed versions of the corresponding estimators, like graph / manifold regularized SVM.
> >
> > We thank the reviewer for sharing their comments about the relation to other algorithms. As stated earlier, we made a conscious choice to exclude empirical analysis of these other algorithms to not give the impression that we are trying to add to that framework or “beat” benchmarks. Nevertheless, we have commented on connections to these algorithms and if the reviewer believes this is essential, then we can include some empirical comparison to these algorithms in the appendix.
> >
> > Minor comments:
> > > In the introduction, one type of self supervised learning missing is the other major class of SSL not mentioned - predicting held-out parts of the input data (as in transformer models)
> >
> > We have added this remark to the updated draft as requested.
> >
> > > Figure 1 does not seem very helpful to me and doesn't seem to explain anything.
> >
> > We believe such a figure is helpful to people who are unfamiliar with kernel methods to see the mapping between the standard setting and that of the kernel setting. Perhaps to help us better understand why the figure is not helpful, can the reviewer expand on this point?
> >
> > (2/2)

---

### Decision · Program_Chairs · 2023-01-20

**Decision:**

Reject

**Justification For Why Not Higher Score:**

The concerns of not having substantive contributions vis-vis existing work is not effectively addressed

**Justification For Why Not Lower Score:**

N/A

**Metareview: Summary, Strengths And Weaknesses:**

This paper takes a Kernel learning approach to Self Supervised Learning. It provides a novel representation and the associated
dot-product provides a new kernel. It provides some analysis of this kernel on downstream applications.
The main weakness seems to be that in light of the substantial body of work on kernel methods for graphs spanning two decades the contributions seem to be modest. Experimental evaluation also seems to be not too comprehensive.
It is thus clear that the paper needs to be improved to appear in a premier venue.



**Summary Of Ac-Reviewer Meeting:**

Not needed